# Advancing macromolecular structure determination with microsecond X-ray pulses at a 4th generation synchrotron
Julien Orlans[1], Samuel L. Rose [1], Gavin Ferguson [2], Marcus Oscarsson[1], Alejandro Homs Puron[1], Antonia Beteva[1], Samuel Debionne[1], Pascal Theveneau[1], Nicolas Coquelle[1], Jerome Kieffer [1], Paolo Busca[1], Jeremy Sinoir[3], Victor Armijo [3], Marcos Lopez Marrero[3], Franck Felisaz[3], Gergely Papp [3], Herve Gonzalez[1], Hugo Caserotto[1], Fabien Dobias[1], Jonathan Gigmes[1], Guillaume Lebon [2], Shibom Basu [3] ✉ & Daniele de Sanctis [1] ✉

Serial macromolecular crystallography has become a powerful method to reveal room temperature structures of biological macromolecules and perform time-resolved studies. ID29, a flagship beamline of the ESRF 4th generation synchrotron, is the first synchrotron beamline in the world capable of delivering high brilliance microsecond X-ray pulses at high repetition rate for the structure determination of biological macromolecules at room temperature. The cardinal combination of microsecond exposure times, innovative beam characteristics and adaptable sample environment provides high quality complete data, even from an exceptionally small amount of crystalline material, enabling what we collectively term serial microsecond crystallography (SμX). After validating the use of different sample delivery methods with various model systems, we applied SμX to an integral membrane receptor, where only a few thousands diffraction images were sufficient to obtain a fully interpretable electron density map for the antagonist istradefylline-bound $A_{2A}$ receptor conformation, providing access to the antagonist binding mode. SμX, as demonstrated at ID29, will quickly find its broad applicability at upcoming 4th generation synchrotron sources worldwide and opens a new frontier in time-resolved SμX.

The advent of serial femtosecond crystallography at X-ray free electron lasers (XFELs) pioneered the development of new methods for the determination of macromolecular structures. The high brilliance of the source generates a photon flux density so intense that it evaporates the microcrystalline samples upon exposure, yet produces, under favourable conditions, an interpretable diffraction pattern[1–3]. The collection of diffraction data from hundreds to thousands of single-shot diffraction patterns (i.e. without crystal rotation) allows the reciprocal space to be reconstructed and the corresponding electron density map to be calculated. This approach revived the interest for structures determined at room temperature and its importance for identifying physiologically relevant conformations, and their time dependance, which could be hindered by using conventional cryo-crystallography, thus, allowing the observation of biological and biochemical reactions that could occur in crystals[4–9]. A typical experiment at XFELs often requires a large amount of sample, a complex experimental setup and limited access

to the end-station, which is a major bottleneck to expand the application into the broader domain of structural biology[10].

At 3rd generation synchrotrons, the arrival of faster photon counting detectors[11,12] and technology transfer from XFELs[13,14] enabled rapid data collection with < 10 ms exposure time. This in turn facilitated so-called room temperature serial millisecond crystallography (SMX)[15]. However, unlike XFELs, where radiation damage is outrun thanks to the femtosecond X-ray pulses, SMX at 3rd generation synchrotron relies on data collection at a very low-dose per crystal[2,14,16], limiting in some cases, the highest achievable resolution. Conversely, limited photon flux density and achievable detector exposure times restricts the achievable time resolution to a few milliseconds.

An alternative approach has consisted in exploiting the so-called "pink-beam", where a partially polychromatic beam can be used to produce highly intense X-ray pulses often from spatially separated bunches of electrons in the storage ring, reaching in this way a time resolution of hundreds

[1]ESRF - The European Synchrotron, 71 Avenue des Martyrs, Grenoble, France. [2]Institut de Génomique Fonctionnelle, Université de Montpellier, CNRS, INSERM, Montpellier, France. [3]European Molecular Biology Laboratory, 71 Avenue des Martyrs, Grenoble, France. ✉e-mail: shbasu@embl.fr; daniele.de_sanctis@esrf.fr

of picoseconds. The approach of Laue crystallography has been successful in the pioneering study of photoreversible photoinduced structural changes and it has been more recently applied in serial crystallography (SX) experiments either using fixed target (albeit with crystals larger than 10 μm) or High Viscosity Extruders (HVEs), using a larger spectral distribution than conventional MX beamlines (from 2.4% to 5% $\Delta E/E$)[17-19] and a beamsize (much) larger than state of the art microfocus beamlines. Thus, the major challenge is to find an appropriate balance between temporal and spatial resolution to track biological reactions in crystals. While this approach is capable of recording more reciprocal space information out of a single shot at high time resolution, it is intrinsically more sensitive to background level and spatial resolution, at the detriment of the achievable signal-to-noise. Moreover, because of the spectral band, treatment of Laue diffraction patterns requires customised software for data reduction[20]. Therefore, these represent a further bottleneck in applying Laue method to large macromolecules and weakly diffracting targets.

In response to this current challenge, as part of the Extremely Brilliant Source (EBS) upgrade of the European Synchrotron (ESRF), a pioneering microfocus beamline, namely ID29, dedicated to room temperature serial crystallography was entirely designed and built to replace the previous Multiwavelength Anomalous Dispersion (MAD) beamline[21]. The ESRF-EBS[22] implemented a novel concept of a Hybrid Multi Bend Achromat (HMBA) lattice making it the first high energy (6 GeV) 4th generation synchrotron. The unique beamline layout has been devised to exploit the features of EBS, making it capable of delivering a flux density which is two to three orders of magnitude higher than other microfocus beamlines at 3rd and 4th generation synchrotrons. This puts ID29 at the forefront, being able to provide still diffraction patterns from microcrystals with Bragg intensities well above the background, which are produced from a slightly polychromatic (1% $\Delta E/E$) pulsed beam with high repetition rate in the microsecond time regime, overcoming the bottlenecks previously described. This development is of potential interest for other 4th generation synchrotron sources - such as MAXIV, SIRIUS, APS-U, SLS2.0, Diamond-II, ELETTRA 2.0, Petra-IV, SPring-8-II, HEPS and ALBA II - which are in operation, under development or in the project phase.

ID29 commenced its user operation at the end of 2022 and had been used for data collection on a wide variety of samples and SX experiments. Here we describe the first comprehensive application of serial microsecond crystallography (SμX) at the new ESRF-EBS beamline, using sample delivery methods already established in SX - fixed target, either on foils or Silicon (Si-) chips, and HVEs. This approach opens new perspectives to investigate time-resolved studies that are out of reach at 3rd generation sources, and for which femtosecond time scales are superfluous. SμX was used to determine the room temperature structures of three classic targets (proteinase K, thaumatin, and the lysozyme-GlcNAc complex) and a human integral membrane protein, which belongs to the large family of G protein-coupled receptors (the adenosine receptor $A_{2A}R$ - co-crystallised with Istradefylline). This selective $A_{2A}R$ antagonist is used as an adjunctive treatment for the management of "off" episodes associated with long-term treatment of Parkinson's disease with levodopa/carbidopa[23]. The experiments presented here reveal that the recently developed data collection environment and the ID29 beam characteristics, allow for a routine and efficient application of room temperature SμX with an optimal usage of the crystalline material to obtain complete and high multiplicity data in a few minutes. Moreover, the microsecond pulse-length paves the way for a standardised application of time-resolved studies down to this time scale. This will expand the use of serial crystallography at synchrotrons to a comparable throughput and efficacy as conventional rotation crystallography, and eventually to broad availability to other future sources.

## Results
### Serial microsecond data collection at ID29
ID29 presents a completely new and innovative design, specifically conceived to exploit the characteristic of the EBS lattice. Briefly, the layout has been designed to preserve most of the brilliance of the X-ray source and

deliver a photon flux density of three to four orders of magnitude higher than 3rd generation sources' beamlines at the sample position. Similar to pink-beam experiments at Laue beamlines[24,25], and unlike serial millisecond crystallography at 3rd generation sources, the X-ray beam is mechanically pulsed, thanks to a chopper system. A "less monochromatic" beam, with 1% $\Delta E/E$ bandwidth delivers a photon flux of $\sim 2 \times 10^{15}$ ph/s in a $4 \times 2\ \mu m^2$ focal spot (Fig. 1a). In this work, the beam was divided into routine pulses of 90 μs, at a repetition rate of 231.25 Hz (1/4 of the main operation frequency of the beamline at 925 Hz). We note that the operation with shorter pulses, down to 10 μs, achievable with an additional second chopper, is currently being commissioned. These experimental conditions are today unrivalled and are leading to a new direction in the use of X-ray diffraction for structural studies at room temperature. Diffraction data were recorded with a charge-integrating JUNGFRAU 4 M detector[26], automatically corrected and geometrically reconstructed with an in-house developed data acquisition system, LImA2[27] (Fig. 1b). Finally, a new diffractometer, the MD3upSSX, was developed specifically for ID29, as an evolution of the MD3up, with specific features for serial crystallography. The MD3upSSX introduces the possibility to enslave the data acquisition sequence to an external clock (i.e. matching the frequency of the pulsed beam), to which the whole experiment is synchronised, and includes the possibility to trigger external devices, such as laser sources for pump-and-probe experiments[28], with configurable time delays. Our diffractometer is part of a versatile experimental setup which offers two main configurations: fixed target and injector data collection mode (Fig. 1a, Supplementary Fig. 1), both of which are interchangeable in less than 1 h (Fig. 1). In either configuration, the experiment is fully synchronised on the arrival of the pulses from the chopper with the detector triggered for the acquisition of each frame.

**Software to assist SμX data collection.** To support SμX data collections at ID29, a dedicated version of *MXCuBE-Web*[29]—based on the latest development of *mxcubecore* (https://github.com/mxcube/mxcubecore) —was developed. *MXCuBE-Web* for SμX preserves the same intuitive and powerful interface of its MX precursors, and facilitates different types of SX data collections, including (but not limited to) fixed-targets or HVEs. The data acquisition pipeline has been specifically developed using the LImA2 framework for distributed detectors. LImA2 embeds an on-line data analysis plugin, originally inspired by NanoPeakCell[30], to identify crystal hit-finding by harnessing GPU-based implementation of Peakfinder8 algorithm[31] within pyFAI suite[32]. A frame is considered a hit if it contains a significant number of diffraction spots, spread across multiple contiguous pixels above a predefined threshold, with a large enough signal-to-noise ratio. Maximum intensity projections are generated from all frames labelled either as a "hit" or as "non-hit" for comparison, with the option of discarding the latter (Fig. 1b).

### SμX on High Viscosity Extruders
The MD3upSSX diffractometer, as shown in Fig. 1, is complemented with a motorised 3-axis translational stage which is able to accommodate and align any type of injector-style sample-delivery system to the beam position. In this work, three different types of HVEs—developed by the Arizona State University (ASU-HVE)[13], Max-Planck Institute (MPI-HVE)[33], and RIKEN SPring-8 Center (SACLA) (SACLA-HVE)[34]—were used to demonstrate the ease of operation of all three and the versatility of the ID29 experimental setup (Supplementary Figs. 1d–g). Thaumatin microcrystals embedded in Hydroxyethyl-cellulose (HEC) matrix were extruded through a fused silica capillary of 100 μm inner-diameter. The extrusion speed was adjusted in each device in the range of 2–3 μm/ms, with the medium moving by ~10 μm between two consecutive X-ray pulses to ensure that the sample is fully refreshed. At this extrusion speed the sample moves by less than 0.3 μm during the 90 μs exposure (Supplementary Table 1), ensuring that the crystal is constantly in the beam during this time and maintaining a minimal background of maximum 2 photons (Fig. 2a, b and Supplementary Fig. 2). This allowed 500k images to be collected in approximately 35 min for each type of HVE, with a reasonably high hit-rate (from 11.6% to 26.9% with

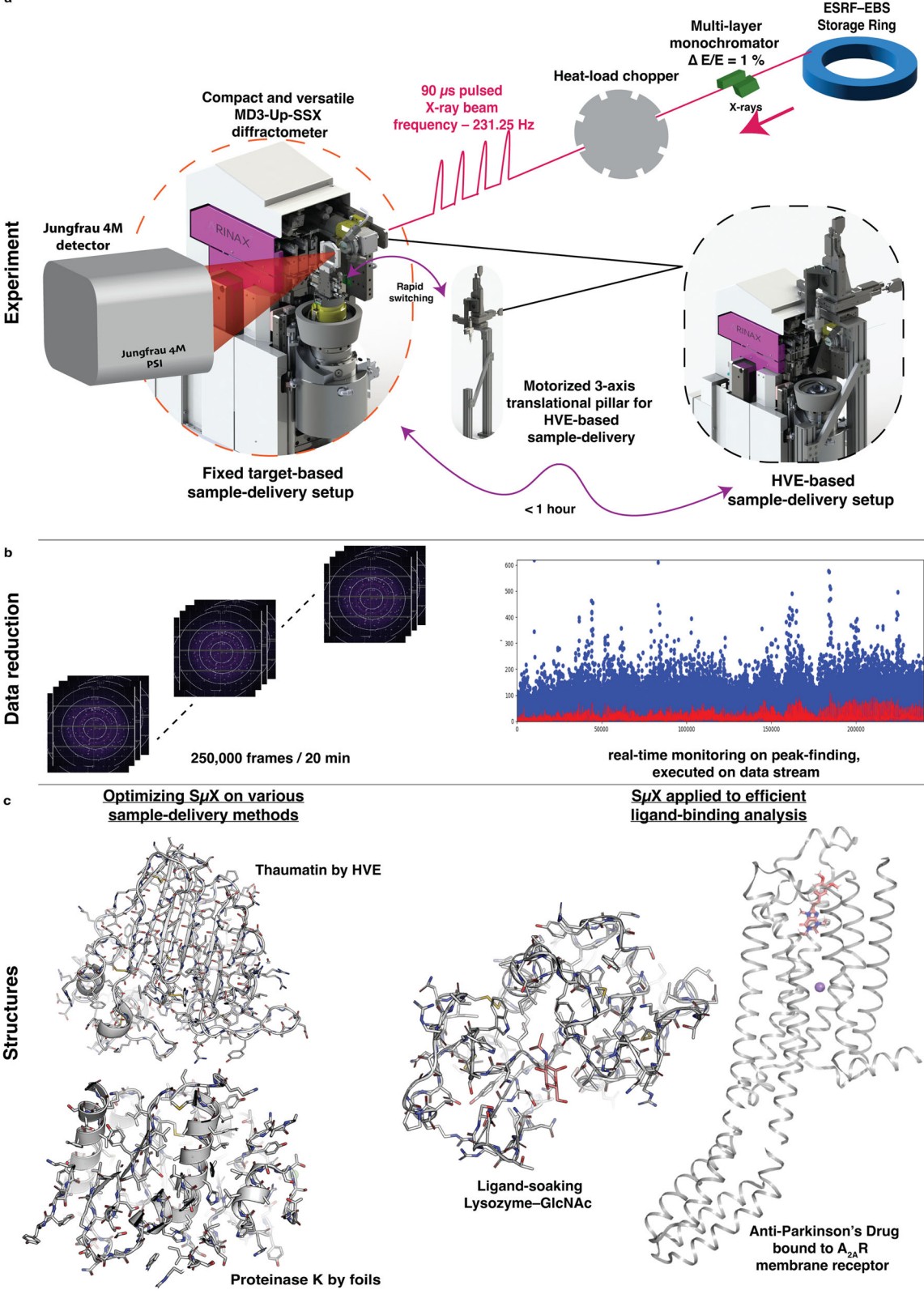

**Fig. 1 | Schematics to show the entire SµX pipeline at ID29. a** The setup integrates a chopper producing 90 µs X-ray pulses, focused to 4 × 2 µm², a compact/versatile MD3upSSX diffractometer, enabling fast switching between different sample delivery tools and a JUNGFRAU 4 M detector. Diffractometer scan and detector acquisition are fully synchronised with X-ray pulses at 231.25 Hz frequency. Switching between fixed-target and HVE setups takes < 1 hr and is being routinely done for users during a single beamtime session. **b** A real-time data reduction software has been integrated into data collection setup to rapidly identify crystal hits. **c** SµX on various delivery methods yielded high resolution structures, that includes identifying GlcNAc ligand, soaked in lysozyme crystals and Istradefylline, an anti-parkinson's drug, co-crystallized with a human integral membrane protein $A_{2A}R$ receptor. The structures are represented as cartoons in grey and ligands are highlighted in sticks (salmon-red), made with PyMOL[94].

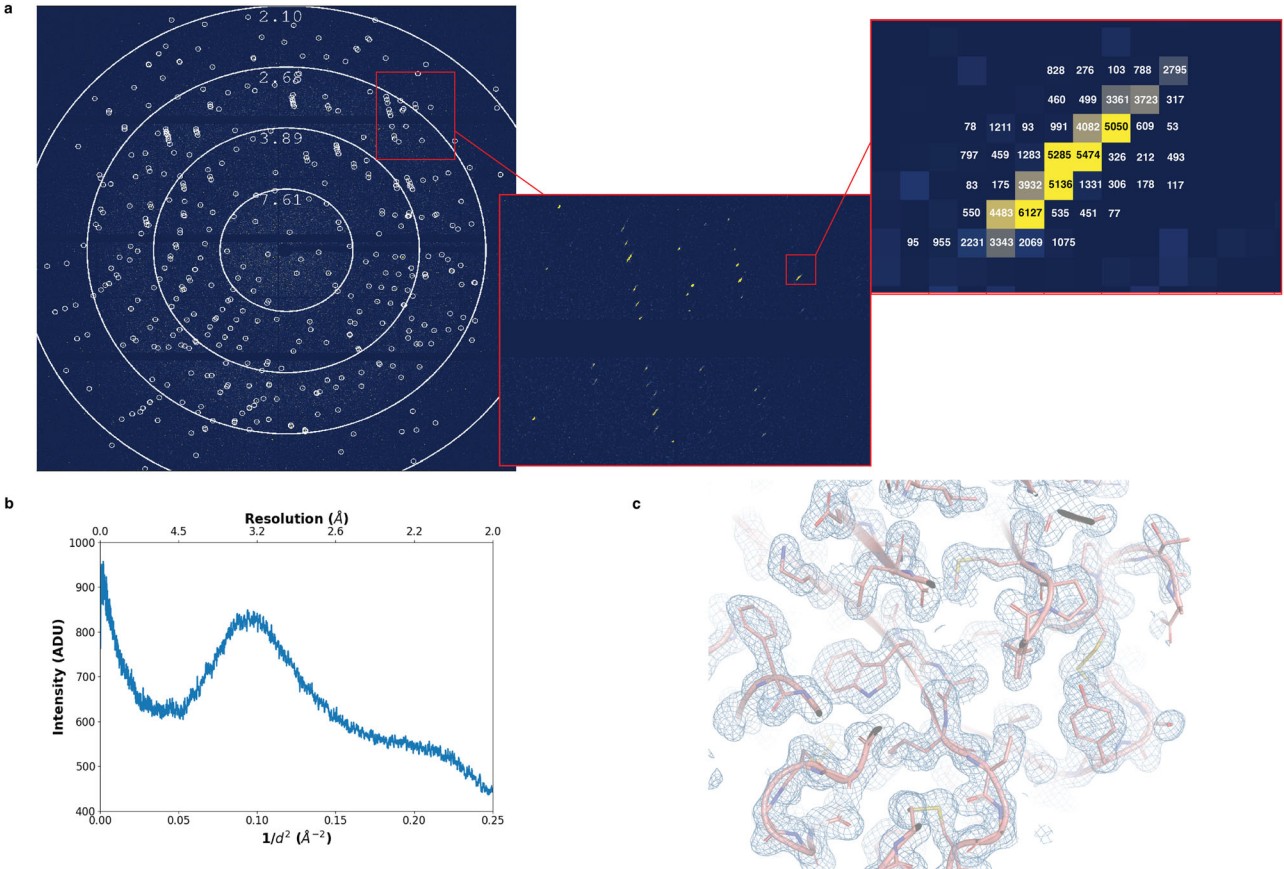

**Fig. 2 | Diffraction intensities and electron density map obtained from a SµX experiment using HVE. a** Single frame diffraction image from a still crystal. Detector integration time is set to 95 µs for a 90 µs pulse. Identified Bragg peaks are marked by circles for better visualisation. Insets show the elongated shape of the Bragg spots, because of beam divergence and bandwidth, and the low background compared to peak intensities. Units are in Analog-Digital-Units (ADUs, 1 photon corresponding to 478 ADU). **b** The background was quantified by the radial average of a no-hit frame and plotted as a function of scattering vector, $1/d^2$. **c** $2Fo$-$Fc$ electron density map (light-blue) of thaumatin (salmon) contoured at 1.0 $\sigma$ and overlaid with the refined model (figure done with PyMOL[94]).

minimal variation due to differences in crystal embedding in HEC), producing structures of similar quality at 1.75 Å resolution (Table 1, Fig. 2c, Supplementary Table 2).

## SµX on Fixed-targets

Acquisition on fixed targets is performed by continuously scanning in a S-shape trajectory across a predefined area of any permissible dimension (Supplementary Fig. 1a–c). The new sample head of the MD3upSSX (SSX head) allows for rapid raster scan data collection (up to a speed of 125 mm/s) with the translation fully synchronous with the X-ray pulses. The definition of the support geometry adapts the standard descriptor[35] with an aim to promote inter-facility exchanges and easy adoption of newly developed chips. To demonstrate the ease of operation of fixed-target collection on ID29, we used the Si-based Oxford chips[35,36] and mylar-film based sheet-on-sheet (SOS) sandwich[37]—which we refer to as large and small foils— mounted on either 30 × 30 mm² or 8 × 4 mm² frames, respectively. Si-chips present a defined geometry, where microcrystals are trapped into < 10 µm funnel-shaped apertures that are spaced by 125 µm[35,36,38–40] (Supplementary Fig. 1c). The MD3upSSX integrates a calibration feature, similar to what is described elsewhere[40–42], which identifies three fiducials[35] to orient the chip and automatically aligns each "city block" prior the data collection. At the repetition rate of 231.25 Hz, the scanning speed for Oxford Si-chips is 28.93 mm/s, with the chip moving only 2.6 µm within a single pulse (Supplementary Table 1). This is smaller than the size of the aperture and is comparable with the beam size. For foils, the only geometrical description is the scannable area, which is predefined for the support type and can be

specifically modified for custom devices developed by users (Supplementary Fig. 1a, b). Given the high reproducibility of the positioning on the SSX head base, no pre-alignment is needed, the only prerequisite being the horizontal and vertical spacings between X-ray pulses, from which the total number of images to be collected is calculated. On foils, the scanning speed is automatically defined from the given spacing. For example, with a 10 µm spacing, the scanning speed is of 2.31 mm/s, for a movement of only 0.2 µm during the X-ray pulse exposure (Supplementary Table 1).

We investigated the effect of spacing on data quality, using proteinase K microcrystals with a spacing of 10 µm and 20 µm, on small foils, and 50 µm, on large foils. The analysis of integrated intensities (counts in photons), normalised over 7000 indexed diffraction frames as a function of resolution or scattering vector, shows that the integrated intensities are equivalent for different spacing, indicating that there is no intensity decay for the smaller spacing (Fig. 3a). Moreover, despite the difference in the spacing, all data presented very similar hits and indexing rates (3.2–6.1%), with the 20 µm spacing presenting slightly lower B-factor and higher diffracting power, possibly representing the best option for optimal data collection. The respective structures could be refined at 2.0 Å resolution (Table 1 and Supplementary Table 3) and present no relevant conformational differences with similar overall and site specific (such as disulphide bonds and calcium ions) B-factor values (Fig. 3b, c).

## SµX efficiently highlights ligand electron densities

**HEWL-GlcNAc**. An important application of room temperature SX is to highlight ligand binding near to physiological conditions and its impact

## Table 1 | Summary of data quality from SµX experiments

| Datasets | N. images | Overall indexing-rate (%)[a] | Max Res (Å) | CC* (%) | Multiplicity | Rwork / Rfree (%) | Sample consumption | Effective data collection time |
|---|---|---|---|---|---|---|---|---|
| HEWL apo Si-chip | 51,200 | 12.15 | 2.05 | 99.58 | 101 | 21.46 / 25.03 | 2 Si-Chips (100–200 µL each) | ~20 min (~10 min / chip with alignment) |
| HEWL - GlcNAc Si-chip | 51,200 | 14.13 | 2.05 | 99.63 | 123 | 20.67 / 26.50 | 2 Si-Chips (100–200 µL each) | ~20 min (~10 min / chip with alignment) |
| Proteinase-K foils 10 µm | 327,600 | 5.22 | 2.00 | 98.40 | 221 | 21.84 / 26.24 | 1 small foils (3 µL) | ~25 min |
| Proteinase-K foils 20 µm | 163,600 | 6.12 | 2.00 | 96.53 | 132 | 24.04 / 26.24 | 2 small foils (3 µL each) | ~12 min (~6 min / small foil) |
| Proteinase-K foils 50 µm | 250,000 | 3.24 | 2.00 | 96.51 | 99 | 18.22 / 22.38 | 1 big foils (50 µL) | ~20 min |
| A₂ₐR - Istradefylline foils | 435,708 | 3.97 | 2.50 | 98.61 | 209 | 25.08 / 28.77 | 3 small foils (3 µL each) | ~30 min (~10 min / small foil) |
| Thaumatin ASU | 500,000 | 22.84 | 1.75 | 99.87 | 1420 | 13.80 / 16.67 | 12.5 µL microcrystals pellet | ~35 min |
| Thaumatin SACLA | 500,000 | 11.67 | 1.75 | 99.58 | 595 | 15.17 / 18.17 | 12.5 µL microcrystals pellet | ~35 min |
| Thaumatin MPI | 500,000 | 27.68 | 1.75 | 99.93 | 2001 | 14.22 / 17.17 | 12.5 µL microcrystals pellet | ~35 min |

[a]Overall indexing rate calculated for all collected images.

for drug-discovery projects both for academic and proprietary research[39]. As a proof of concept, we have recorded diffraction data from two types of lysozyme microcrystals—apo and pre-soaked with N-Acetylglucosamine (GlcNAc) at 50 mM final concentration, from only two Oxford Si-chips for each crystal preparation at 2.05 Å resolution (Fig. 4a, Table 1 and Supplementary Table 3). The resolution at the time of data collection was limited by the geometrical constraints with a fixed detector distance. This limitation was eventually addressed making it possible to achieve a resolution of 1.6 Å at the edge at 11.56 keV. An isomorphous difference map was calculated to reveal the soaked GlcNAc ligand which could be built in the final model near full occupancy (87%), even at $K_d$ range (47.6 mM)[43]. To verify the viability of this approach in obtaining a clearly interpretable ligand electron density map, we performed a "data titration"[39,44] by reducing the number of merged frames and analysing the $Fo-Fc$ map in the binding pocket (Fig. 4c and Supplementary Table 4). While as expected the density becomes clearer with higher number of frames, we observed that as little as 2,000 frames were sufficient to obtain a fully interpretable electron density for the ligand. To reduce bias from prior knowledge, the ligand was automatically fitted in the $Fo-Fc$ density map from the 2,000 merged images using *phenix.ligandfit*[45], but was rotated by 180° along the glucose C2-C5 direction (Supplementary Fig. 3a). This observation could be justified by the non-planar chair conformation of pyranose of the GlcNAc molecule, which might not allow the software to automatically build it in a correct orientation. As a result, the root mean square deviation (RMSD) between the automatically fitted and the final refined position was slightly higher than expected (2.53 Å, calculated with the webserver *DockRMSD*[46]).

**Adenosine receptor (A₂ₐR)—Istradefylline.** The A₂ₐR receptor is involved in a variety of physiopathological and neurological disorders, such as Parkinson's disease, Pain, inflammation and cancer and represents, consequently, an important target for drug discovery[47,48]. In this study, the A₂ₐR receptor was co-crystallized with the FDA-approved adjunctive drug Istradefylline[49,50] using the Lipidic Cubic Phase (LCP) method[51,52]. GPCR receptors are usually crystallised in LCP to take direct advantage of using the microcrystal-containing matrix as carriers for HVEs. While this approach reduces sample manipulation and has been successfully used both at XFELs and synchrotrons[15], it still requires a large amount of sample. Thus, in order to reduce sample consumption, we exploited the use of foils (small to be specific) and dispensed the content of a crystallisation syringe by gently smearing on the mylar foil before sealing. To maximise the hit rate, we used a 15 µm spacing to ensure a separation larger than the beam and the crystal size (Supplementary Fig. 4). With only 15 µL of crystal slurry embedded in LCP, pooled from three syringes, we could get a complete dataset at 2.5 Å resolution (Table 1 and Supplementary Table 3) with a well-defined electron density around the ligand (Fig. 4b). The "data titration" analysis shows that, even with a lower symmetry space group, just 2,000 merged images (less than half of the frames that was previously reported[53] to be the minimum requirement for interpretable electron densities in SX) were again sufficient to obtain a clear electron density map for the Istradefylline ligand which is refined inside the A₂ₐR pocket at full occupancy (Fig. 4c). The resolution allowed to performed automated docking and the ligand was placed automatically using *phenix.ligandfit* with a RMSD of 0.26 Å from the refined position (Supplementary Fig. 3b), calculated with *DockRMSD*[46]. Istradefylline binds in a pocket similar to the antagonist ZM241385[54]. Istradefylline is stabilised by F168 and L249 through hydrophobic interaction with the xanthine moiety, and polar interaction with N253, a key residue for agonist and antagonist binding[55] (Fig. 5). The binding pocket extends to the extracellular side of the receptor where istradefylline makes contact with H264, E169 and Y271. The binding mode determined here at room temperature agrees with pharmacological and biochemical studies of A₂ₐR bound to Istradefylline[56] and with a high-resolution structure determined at cryogenic conditions (manuscript in preparation).

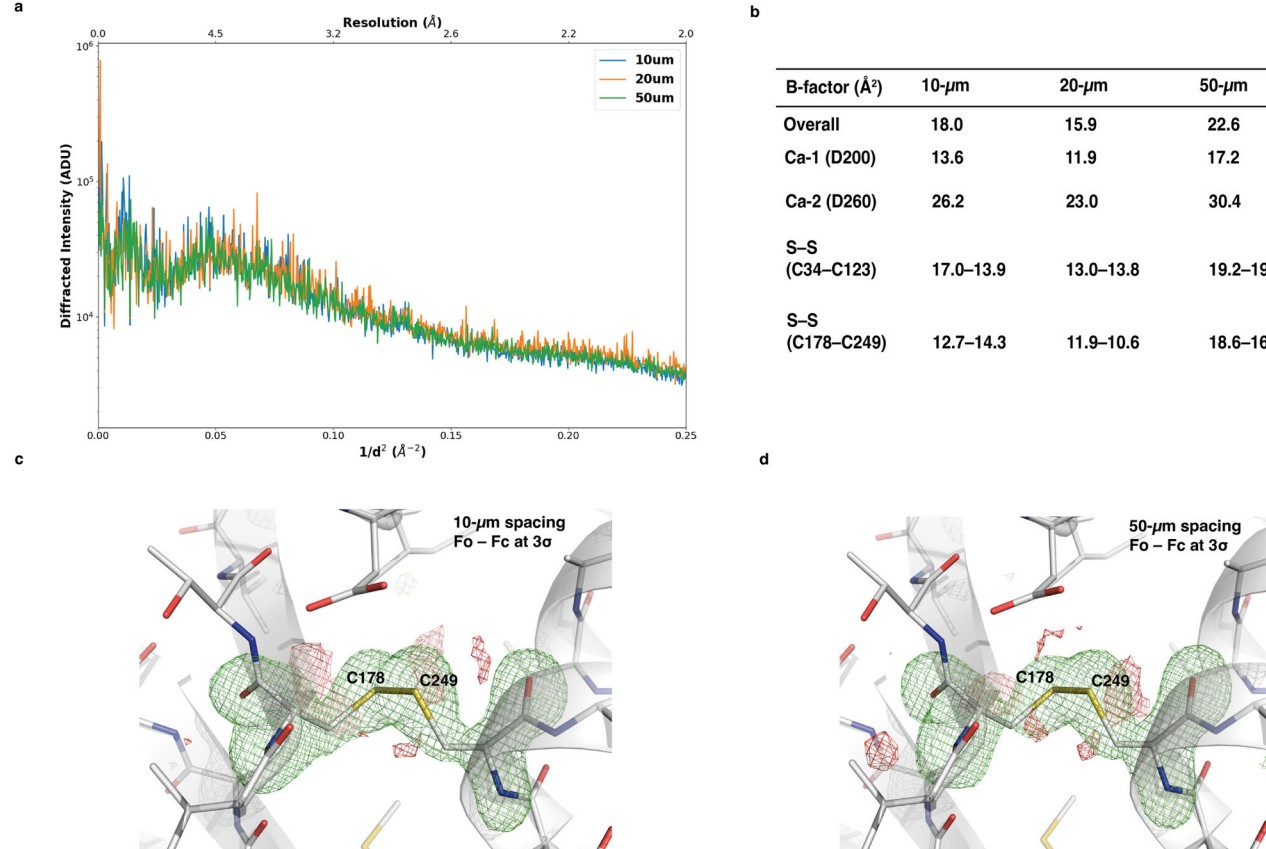

**Fig. 3 | Effects of variable spacing between two X-ray shots in foils SμX. a** Radial distribution of the Bragg peaks integrated intensities (in logarithm scale) as function of resolution calculated from 7000 indexed frames collected on foils at different spacings, 10, 20 and 50 μm. The intensity unit is in Analog-to-digital units (ADU), where 1 photon corresponds to 478 ADU. **b** Overall and site specific temperature factors of proteinase K for 10, 20 and 50 μm spacings. **c, d** *Fo-Fc* omit map (green) at 3.0 σ on C178-C249 disulphide bridge (yellow stick) of proteinase K (shown in grey cartoon) from data sets collected with 10 and 50 μm spacing, respectively.

## Discussion

This work demonstrates the innovative perspectives that 4th generation synchrotron sources provide to exploit serial microsecond crystallography. The characteristics of the ESRF-EBS led to the construction of a new beamline (ID29) which can overcome most of the constraints of 3rd generation synchrotron endstations. The results obtained with our data collection setup and unique beam parameters are unprecedented, representing an innovative use of synchrotron radiation at diffraction limited sources. The data presented here lead the way for the construction and commissioning of similar beamlines at other synchrotrons, currently undergoing or planning the upgrade of their accelerators. The use of a larger bandwidth compared to conventional microfocus MX beamlines allows a higher flux to be conveyed to the sample, thus reducing the exposure time of the data acquisition to tens of microseconds, but also improves the completeness of the spots recorded from still crystals[17,18,57]. In theory, the same concept could also be applied to beamlines at 3rd generation synchrotrons, albeit with longer pulses because of the lower brilliance of their sources. Additionally, the high demagnification produced by the beamline optics introduces a high divergence in the beam on the sample, which, in serial crystallography, has been suggested to be advantageous[58]. Our results here show that a clearly defined electron density map can be achieved with a limited amount of frames, smaller than what is reported for similar experiments[53].

The use of the very short pulses (90 μs) achievable at ID29 brings a distinct advantage compared to 3rd generation synchrotrons. During data collection with HVEs, the microcrystals barely move within a single 90 μs pulse, maximising the signal-to-noise ratio in still diffraction snapshots (Fig. 2 and Supplementary Table 1 and Supplementary Fig. 2). When working at similar extrusion speeds with longer exposures subjected to 3rd

generation synchrotrons, in the range of a few milliseconds, the crystal usually stays in the beam for only a fraction of the exposure time, while background from the carrier medium is recorded during the whole acquisition. The same kind of situation is occurring in the case of data collection on continuously translating meshes or foils. By applying microsecond pulses, the crystalline material remains in the beam for the whole pulse length, maximising the signal-to-noise in the recorded frame by reducing the background. Moreover, the pulsed beam allows the areas exposed to X-ray to be spaced out. This seems to play an important role in the mitigation of radiation damage, ensuring that each diffraction image is collected from an unexposed area. The uniformity in the intensity distribution (Fig. 3a) may indicate that the photoelectron and radical diffusion length is much smaller than the linear spacing between two consecutive X-ray pulses[59], and that no cross-contamination is occurring even with the smaller spacing, unlike at conventional microfocus beamlines with a continuous exposure of the support and the mother liquor.

The flux density achievable at ID29 ( > $10^{14}$ ph/s/μm$^2$) is currently unique, with it being nine orders of magnitude lower than XFELs but at least two to three orders higher than 3rd generation synchrotrons and other MX beamlines at next generation light sources[60,61]. As such, the calculated dose for the experiments in this work are much higher than the ones typically reported for room temperature serial measurements[62] (Supplementary Tables 2, 3 and Supplementary Table 5). While some speculations could be drawn based on previous studies[8,14] the effect of radiation in room temperature crystallography is still not as well characterised as for cryo experiments[62]. The study of the potential benefit of a high-dose rate[63,64], together with a pulsed beam, is of direct interest to drive the design of future serial crystallography beamlines and

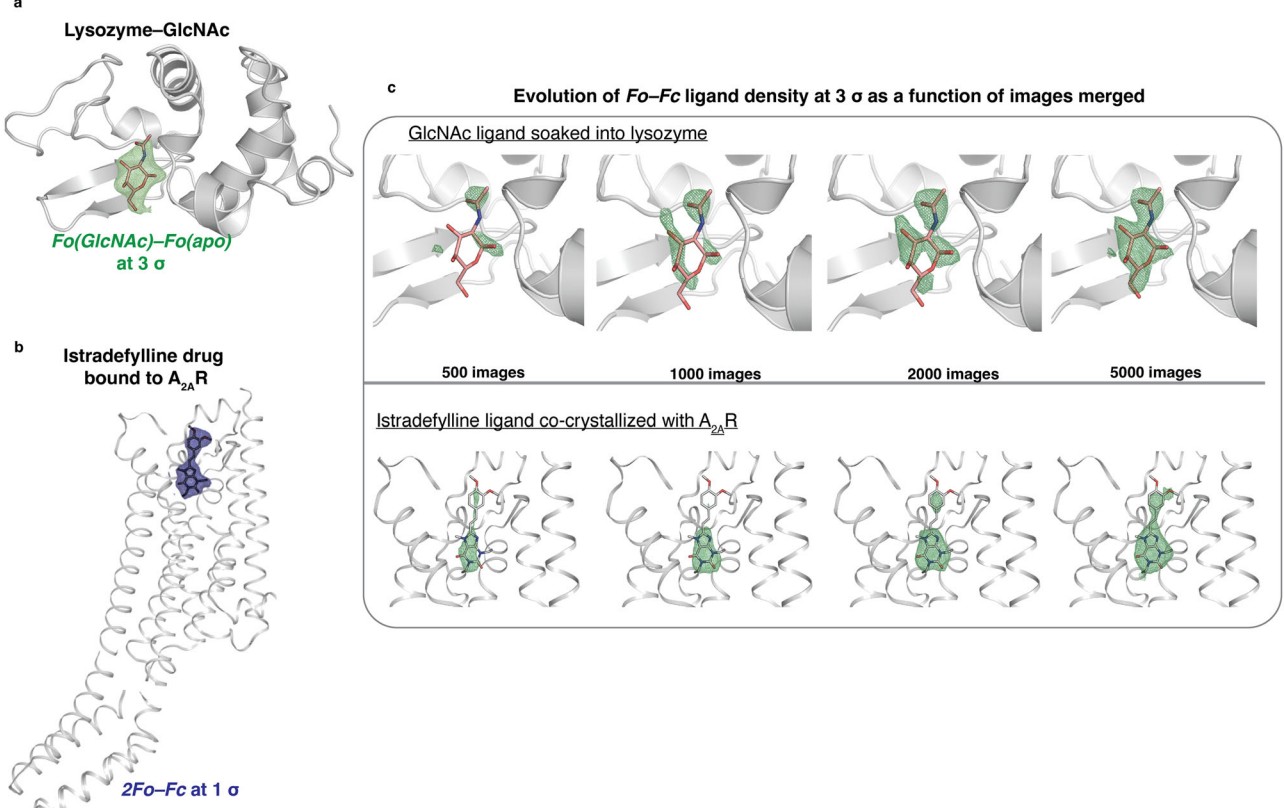

**Fig. 4 | Ligand binding "data titration" analysis. a** Isomorphous difference map – *Fo(GlcNAc)-Fo(apo)* contoured at 3 σ and in green mesh at the resolution of 2.05 Å between ligand-bound and apo lysozyme crystal structure revealing the presence of N-Acetylglucosamine (GlcNAc). **b** *2Fo–Fc* density map (blue mesh) carved around the Istradefylline ligand for the $A_{2A}R$ receptor is shown at 1 σ at the resolution of 2.5 Å. **c** Evolution of (*Fo–Fc*) electron density map (green) contoured around the ligand at 3.0 σ calculated for an increasing number of 500, 1000, 2000, and 5000 merged microcrystals of lysozyme-GlcNAc (top) and $A_{2A}R$-Istradefylline (bottom) at the same resolution cutoff of 2.05 and 2.5 Å respectively.

experiments. The sample environment set-up designed at the ID29 beamline is both flexible and versatile, while still compact, allowing for the integration of various sample-delivery technologies (fixed-target and HVEs) for SµX experiments (Fig. 1a and Supplementary Fig. 1) and easily accommodating for a wide range of sample delivery systems, including microfluidics[65,66], tape-drives[67,68], and acoustic levitators[69]. In the case of Oxford chips (or similar chips such as the HARE[38,40] or the MISP[70] chips) the synchronisation of the fast translating motor with the experimental clock (and the X-ray pulses) permits the acquisition of data by continuously scanning at 231.25 Hz, instead of a much slower "stop and collect" sequence[41]. This, combined with the ID29 repetition rate, brings the alignment and acquisition of a complete chip to less than 10 min. The efficiency of the experiments with fixed targets (together with foils) permits to envisage a conventional-MX inspired automation[71] in future with a robot mounting Si-chips or foils from a humidity-controlled storage chamber, followed by automatic data collection and eventual processing. This, for instance, could lead to the development of room temperature fragment or ligand screening at a comparable throughput to conventional MX and eventually open up the application of SµX to tackle structure-based drug discovery projects for pharmaceutical industry and biotechnological applications[6,72,73]. One could envisage reaching the throughput of up to 100 ligand-soaked data sets in a 10 h beamtime with SµX, with the assumption that 5000 merged images with a 10% indexing rate can be collected in about 5 min.

Finally, the production of 90 µs pulses with our chopper system, and the commissioning of an additional second system, capable of further reducing the exposure time down to 10 µs, also provides a unique set-up for time-resolved experiments at a synchrotron. FELs are best suited to observe ultrafast reactions, such as chemical and electron transfer processes[74,75],

thanks to the femtosecond pulses[76]. However, a time resolution on the microseconds is sufficient for most biological and enzymatic processes, for which an average *Kcat* of ~13 s$^{-1}$ has been reported in literature[77–79]. Data collected with XFELs have been reported with a photon flux of $10^{10}$ to $10^{11}$ ph/pulse[80], which is in the same order of magnitude of ID29, albeit on the microsecond time scale, making it possible to obtain the spatial resolution and structural quality similar to what is reported from XFELs[81]. Although exceptions might exist, this evidence brings ID29 to the forefront as a more accessible tool for SµX experiments and provides the entrance door for standardised time-resolved SX experiments for traditional crystallographers, that are performed on photoactivatable molecules and caged compounds or by fast mixing with ligands or substrates and other emerging approaches for studying protein dynamics[82].

## Conclusions

The groundbreaking experimental potential of ID29, built in the context of the ESRF 4th generation synchrotron upgrade, bridges standard MX beamlines and XFELs and establishes serial microsecond crystallography. The cardinal combination of the unique beamline characteristics together with the streamlined experimental setup and real-time data reduction softwares, collectively enables users to rapidly achieve their goals on multiple projects during a single beamtime. New users of serial crystallography and the expanding time-resolved SX community alike will benefit from SµX. On top of this, the throughput of SuX with the ease of operation and effectiveness of data acquisition will be attractive to industrial users to perform large ligand screening campaigns at near physiological conditions. Upcoming SX beamlines at other synchrotrons are already planned to adopt the ID29 concept, which will set new standards across various facilities and establish these methods within the future generation of structural biologists.

**Fig. 5 | Docking of Istradefylline ligand inside the binding pocket of $A_{2A}R$.** The *2Fo–Fc* map carved around Istradefylline (shown in purple sticks) is contoured at 1.0 σ in light-blue colour. Hydrogen bond networks with water molecules (displayed as red spheres) and with neighbouring residues (displayed as grey sticks) are shown with dashed yellow lines. The ligand is anchored with a polar interaction with N253 and stabilised by pi-stacking interaction with F168 and hydrophobic interaction with L249. Water molecules are mediating H-bond between Istradefylline and Y9/H278 residues. Transmembrane helices are represented as cartoons in transparent grey. Figure was prepared in PyMOL[94].

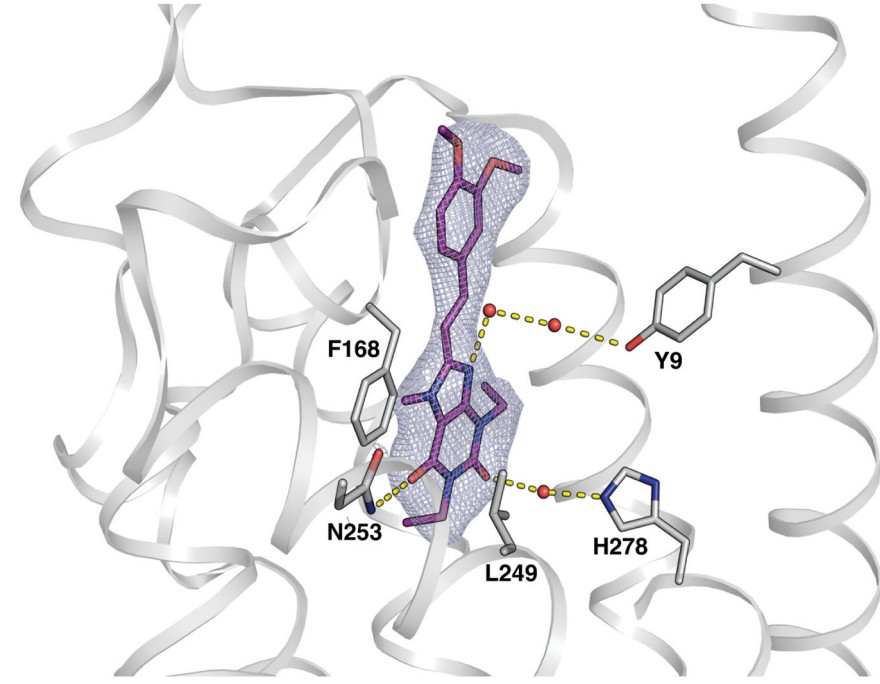

## Methods
### Sample preparation
**Lysozyme**. Hen egg white lysozyme (HEWL) microcrystals were produced using the batch crystallisation method[83]. Briefly, 50 mg/mL HEWL (Sigma #L6876) in 20 mM sodium acetate pH 4.6 was mixed in a 1:1 ratio with 1 M sodium acetate pH 3.0, 20% (w/v) NaCl, 5% (w/v) PEG 6000 and quickly vortexed. Crystals in space group $P4_32_12$ with the dimension of $5 \times 5 \times 5 \ \mu m^3$ appeared almost instantly and rested overnight at room temperature (Supplementary Fig. 4). A volume of 100–200 μL of a ~ $3 \times 10^6$ crystal/mL HEWL microcrystals suspension was dispensed on glow-discharged (PELCO easiGlow™) silicon nitride fixed-target chips, in a humidity-controlled tent (Solo Containment soloLAB™) to avoid sample dehydration. A gentle vacuum suction was applied to remove the excess of liquid before sealing the chip between two Mylar foils of 13 μm thickness (Goodfellow ES30-FM-000230). For ligand soaking, N-acetyl-D-glucosamine (Sigma #A8625) was added to the HEWL microcrystals suspension at a final concentration of 50 mM and incubated for several minutes before loading on the chip.

**Proteinase K**. Proteinase K microcrystals were produced in the batch crystallisation method[84]. Briefly, 46 mg/mL proteinase K (Sigma #P2308) in 0.02 M MES pH 6.5 was mixed in a 1:1 ratio with 0.1 M MES pH 6.5, 0.5 M sodium nitrate, 0.1 M calcium chloride and quickly vortexed. Crystals in space group $P4_32_12$ with the dimension of $4 \times 4 \times 4 \ \mu m^3$ appeared after 24 h of incubation at room temperature (Supplementary Fig. 4). A ~ $2 \times 10^6$ crystal/mL Proteinase K microcrystals solution was sealed between two foils of 13 μm thick Mylar (Goodfellow ES30-FM-000230)[37]. The small foil frame (with a scanning area of $4 \times 8 \ mm^2$) was used to collect 10 and 20 μm spacing between each X-ray pulses, and a big foil (with a scanning area of $30 \times 30 \ mm^2$) was used to collect 50 μm spacing.

**Thaumatin**. Thaumatin microcrystals were produced using the batch crystallisation method[85]. Briefly, 40 mg/mL thaumatin (Sigma #T7638) in 0.1 M HEPES pH 7.0 was mixed in a 1:1 ratio with 0.1 M HEPES pH 7.5, 1.8 M sodium potassium tartrate and quickly vortexed. Crystals in space group $P4_12_12$ with dimensions from $12 \times 12 \times 25$ to $45 \times 45 \times 80 \ \mu m^3$, appeared almost instantly and rested overnight at room temperature (Supplementary Fig. 4). Hydroxyethyl-cellulose (Sigma #09368) was used as a viscous media to carry the Thaumatin microcrystals for the three high viscosity extruders[86]. To prepare a homogeneous sample, 75 μL of 20% (w/v) Hydroxyethyl-cellulose aqueous solution was mixed with 25 μL of dense thaumatin microcrystal suspension with a syringe coupler. For the data collection, ASU (Arizona State University)[13], MPI (Max-Planck-Institut für Medizinische Forschung)[33] and SACLA (TOYAMA Co., Ltd.)[34] injectors were used and loaded with 50 μL of the prepared sample. A sample capillary with an inner diameter of 100 μm was used and the HPLC flow-rate was set in order to allow the jet to move 10 μm between each pulse. To prevent curling of the viscous jet, a He-gas flow of 100 ml_n/min was maintained using a mass flow controller (Bronkhorst EL-FLOW Select #F-201CV-500).

**Adenosine receptor $A_{2A}R$.** The previously reported thermostabilized $A_{2A}$-StaR2-bRIL construct[87] was synthesised (GeneCust) to include a lysine residue at the end of the Flag tag (DYKDDDDK) and cloned into the pFastBac1 plasmid. $A_{2A}$-StaR2-bRIL was expressed in *Spodoptera frugiperda* (Sf9) insect cells (Life technologies) grown in EX-CELL® 420 Serum-Free medium (Sigma #14420 C) using the Bac-to-Bac® baculovirus system (Thermo Fisher Scientific). Insect cell membranes expressing $A_{2A}$-StaR2-bRIL were disrupted by washing and ultracentrifugation in a hypotonic lysis buffer (25 mM HEPES pH 7.4, 10 mM $MgCl_2$, 20 mM KCl, 5 mM EDTA) containing 10 μg/mL DNAse I, 100 μg/mL PMSF, 0.5 μg/mL Leupeptin, 0.7 μg/mL Pepstatin A and a cOmplete™ protease inhibitor cocktail tablet (Sigma #5056489001). Pellets were washed once more with the same hypotonic buffer and an additional wash with a hypertonic buffer (25 mM HEPES pH 7.4, 10 mM $MgCl_2$, 20 mM KCl and 1 M NaCl) with ultracentrifugation steps carried out in between (4 °C for 1 hr at ~200,000 g). Membranes were then resuspended in the hypotonic lysis buffer supplemented with 40% (v/v) glycerol and stored at −80 °C until later use.

Frozen membrane preparations were resuspended into a buffer containing 25 mM HEPES pH 7.4, 400 mM NaCl, 10% (v/v) glycerol, 2 mg/mL iodoacetamide (Sigma #I6125), 10 μM Istradefylline, and a cOmplete™ protease inhibitor cocktail tablet and stirred at 4 °C for 1 hr. Membranes were solubilized by adding a final concentration of 1% (w/v) n-dodecyl-β-D-maltopyranoside (DDM, Anatrace #D310LA), and 0.2% (w/v) cholesteryl hemisuccinate (CHS, Anatrace #CH210) with additional stirring for 1.5 h at 4 °C. The supernatant was collected by ultracentrifugation at ~200,000 g for 1 hr at 4 °C and filtered with low binding

0.22 µm syringe filters. The filtered supernatant was supplemented with 10 mM imidazole before loading onto a pre-equilibrated nickel ($Ni^{2+}$) resin HisTrap™ HP column (GE Healthcare) at 0.5 mL/min overnight at 4 °C on an ÄKTA Purifier system. Once the protein was loaded, the HisTrap™ column was washed using a 4-step gradient of 10 mM, 20 mM, 40 mM and 50 mM imidazole and purified receptor was eluted in a final buffer containing 25 mM HEPES pH 7.4, 200 mM NaCl, 10% (v/v) glycerol, 0.05% (w/v) DDM, 0.01% (w/v) CHS, 250 mM imidazole and 10 µM Istradefylline. The eluted fractions were pooled and concentrated to ~500 µL using an Amicon Ultra-15 centrifugal concentrator MWCO 100 kDa (Millipore), followed by 10 min centrifugation at ~230,000 $g$ to remove aggregates prior to loading on a Superdex 200 Increase 10/300 (GE Healthcare) size exclusion chromatography (SEC) column. The final SEC buffer contained 25 mM HEPES pH 7.4, 150 mM NaCl, 0.03% (w/v) DDM, 0.006% (w/v) CHS, and 10 µM Istradefylline. Fractions containing the Istradefylline-bound receptor were concentrated to 35 mg/mL and flash frozen in liquid nitrogen and stored at −80 °C until crystallization. Purified $A_{2A}$-StaR2-bRIL receptor was mixed with monoolein, 9.9 MAG (Molecular Dimensions #MD2-67), supplemented with 10% (w/w) cholesterol (Anatrace #CH200) using an LCP syringe coupling system for a final protein:lipid ratio of 2:3 (v/v)[52]. For scaling-up the required LCP volumes, $A_{2A}$R crystals were grown in 100 µL Hamilton gas-tight syringes as previously described[88] and incubated for up to one month at 20 °C before beamline experiments. Preparations for loading each crystal-laden LCP sample on foils were carried out under red light immediately before data collection. Briefly, a needle was fastened on the Hamilton gas-tight syringes, and the precipitation solution slowly expelled from the syringe without removing the LCP. After all the precipitation buffer was removed, the LCP was evenly distributed on small foils and carefully spread once the device was sealed. The diffraction data were collected from those small foils with a 15 µm spacing between two consecutive pulses. From the collection of several preparations, the highest number of diffraction quality crystals grew in syringes in space group C222$_1$ and consisted of precipitant solutions of 0.1 M tri-sodium citrate pH 5.0, 50 mM sodium thiocyanate, 29–30% (v/v) PEG 400, and 1% (v/v) 1,6-hexanediol (Sigma #88571).

### Data collection parameters
Measurements were performed at an X-ray energy of 11.56 keV (1.072 Å) with a 1% bandwidth and a beam size of $4 \times 2$ µm$^2$ (H × V) and a maximum flux of $2 \times 10^{15}$ ph/s. 90 µs pulses were generated at a repetition rate of 231.25 Hz. Acquisition was performed with a Jungfrau 4 M detector with integration time of 95 µs to ensure full recording of the X-ray pulse. The sample-to-detector distance was kept constant at 150 mm. Data collection statistics are shown in Supplementary Tables 2, 3.

### Data processing
Diffraction patterns acquired at ID29 beamline are processed using CrystFEL-v0.10.1 suite[89]. Within the CrystFEL suite, spot-finding was performed using *peakfinder8* algorithm[31]. Various indexing algorithms – *xgandalf*[90], *mosflm*[91], *XDS*[92] and *asdf*[93] were applied to find spots and index diffraction frames with the following parameters: --threshold=1500; --min-snr=5; --min-pix-count=2; --peak-radius=4.0,6.0,10.0; --min-peaks=10; --multi; --local-bg-radius=5; --int-radius=4.0,6.0,10.0. Merging was done using *partialator*[93] with the following parameters: --model=unity; --no-pr; --iterations=3. Structures were determined by molecular replacement using Phaser[94]. Initial phases for lysozyme, proteinase K, thaumatin, and $A_{2A}$R were obtained from PDB-IDs – 7BHK[67], 5KXU[95], 5WR8[86] and 5NM2[15], respectively. All structures were refined with *phenix.refine*[96] and models built with COOT[97] using the PHENIX suite[98]. Figures were made using PyMOL[99] (Schrödinger, LLC). The ligand GlcNAc was placed in the HEWL model by examination of the *Fo-Fc* and *Fo(GlcNAc)-Fo(apo)* maps. The ligand Istradefylline was placed in the $A_{2A}$R model using *phenix.ligandfit*[45] in the *Fo-Fc* map. Final refinement statistics for all structures are shown in Supplementary Tables 2, 3.

### Calculation of dose
The absorbed dose (Gy (J/kg)) on a crystal upon interaction with a 90 µs X-ray pulse was calculated using RADDOSE-3D[100]. RADDOSE-XFEL[64] was not used as microsecond time scales are incomparable to the femtosecond time scales assumed for these simulations. Likewise, the Monte Carlo program[59] within RADDOSE was excluded despite offering several benefits for micro- beam and sample sizes. It was omitted as fewer dose metrics are output for analysis and importantly to allow for scenarios for serial collection down the line using multiple wedges and/or offsets, for example, which are not possible within this program. As such, due to the nature of this type of data collection, several caveats had to be taken into account in the RADDOSE-3D simulation to ensure a dose value per crystal that is as accurate as possible. Firstly, due to microsized crystals and the microfocused nature of the beam, any photoelectron escape from the crystal volume following interaction with the X-ray pulse was taken into account for the simulations. Similarly, photoelectron entry into the crystal volume from material surrounding the crystals was also used in calculations. For fixed-target, an additional parameter was used in simulations to take into account the container and material encasing the irradiated sample also i.e. Mylar was also used in the calculations. This is due to the X-ray pulses passing through this material before depositing the dose. For each simulation, an exposure time of 90 µs and a wedge of 0° rotation was used, together with a calculated flux in ph/s. The average dose for the exposed region of the crystal (ADER) and the whole crystal (ADWC) was calculated and used as the output values for the dose per crystal (Supplementary Table 5). Scripts are provided in Supplementary Data 1.

### Reporting summary
Further information on research design is available in the Nature Portfolio Reporting Summary linked to this article.

### Data availability
The raw data collected and generated in this paper are available in the ESRF data portal through the following https://doi.esrf.fr/10.15151/ESRF-DC-2008212598 (Thaumatin using ASU HVE), https://doi.esrf.fr/10.15151/ESRF-DC-2008703156 (Thaumatin using SACLA HVE), https://doi.esrf.fr/10.15151/ESRF-DC-2008703164 (Thaumatin using MPI HVE), https://doi.esrf.fr/10.15151/ESRF-DC-2008703181 (Proteinase K using small foils - 10 µm spacing), https://doi.esrf.fr/10.15151/ESRF-DC-2008703172 (Proteinase K using small foils - 20 µm spacing), https://doi.esrf.fr/10.15151/ESRF-DC-2008703189 (Proteinase K using large foils - 50 µm spacing), https://doi.esrf.fr/10.15151/ESRF-DC-2008703197 (Lysozyme-GlCNAc using Si-chip), https://doi.esrf.fr/10.15151/ESRF-DC-2008212588 (A2AR-Istradefylline using small foils). All crystallographic data have been deposited in the Protein Data Bank (PDB) under accession codes: 9FTU (Thaumatin using ASU HVE), 9FTS (Thaumatin using SACLA HVE), 9FTV (Thaumatin using MPI HVE), 9FTX (Proteinase K using small foils - 10 µm spacing), 9FTY (Proteinase K using small foils - 20 µm spacing), 9FU1 (Proteinase K using large foils - 50 µm spacing), 9FUD (Lysozyme apo using Si chip), 9FUE (Lysozyme-GlCNAc using Si-chip) and 9FUP ($A_{2A}$R-Istradefylline using small foils).

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

## Acknowledgements

We acknowledge beamtime provision at the European Synchrotron Radiation Facility (ESRF) granted from ID29 BAG proposals MX2434, MX2545 and MX2523. This work was developed within the ESRF-EMBL Joint Structural Biology Group research program and with the support of the Grenoble Partnership for Structural Biology. J.O., S.B. and D.d.S. wish to thank Ilme Schlichting, Bruce Doak and Robert Shoeman from Max-Planck-Institut für Medizinische Forschung, Heidelberg for sharing the MPI injector and the small foil frames. This work is one of the first results of the ID29 beamline flagship project EBSL8, which was possible thanks to the continuous support of past and present ESRF management and directors, several groups at the ESRF: Structural Biology, ISDD Software, ISDD Detector and Electronics, and TID Systems and Communications. G.F. and G.L. acknowledge support from the French National Research Agency, ANR Grant PRC SWITCH-ON (ANR-20-CE11-0019). This work is dedicated to the memory of our colleague Nicolas Janvier who greatly contributed in engineering ID29 control electronics.

## Author contributions

J.O., S.B., D.d.S. designed the experiment. J.O. prepared most of the samples and with S.B. and D.d.S. collected all the datasets. J.O. with input from S.B., S.L.R. and D.d.S. analysed most of the structures determined. G.F., G.L. purified and crystallised A$_{2A}$R and prepared the samples for experiment. A.H.P., S.D., N.C., J.K., P.B. developed different parts of the detector control and data acquisition pipelines. M.O. and A.B. developed the experimental control software. H.C., and H.G. developed the chopper synchronisation system. J.G., F.D. and H.C. with the help of J.O., S.B. and D.d.S. assembled the experimental setup. J.S., V.A., F.F., M.M. and G.P. designed and developed the diffractometer hardware and control software. S.L.R. with input from J.O. calculated the absorbed dose. P.T. and D.d.S. designed the experimental setup. J.O., S.B., S.L.R and D.d.S. wrote the manuscript with input from all authors.

## Competing interests

The authors declare no competing interests.
