## [Peer Review file · Communications Chemistry]

Advancing macromolecular structure determination with microsecond X-ray pulses at a 4th generation synchrotron

Corresponding Author: Dr Daniele de Sanctis

This manuscript has been previously reviewed at another journal. This document only contains information relating to versions considered at Communications Chemistry.

Version 0:

Reviewer comments:

Reviewer #1

(Remarks to the Author)

The manuscript from de Sanctis et al. titled "Advancing macromolecular structure determination with microseconds X-ray pulses at a 4th generation synchrotron" presents the development and application of serial microsecond crystallography ($S\mu X$) using high-brilliance, high-repetition-rate X-ray pulses at the newly constructed ID29 beamline at the ESRF-EBS. It introduces innovations in beamline design, sample delivery methods, and real-time data acquisition techniques, with applications to model systems and integral membrane proteins.

The work is scientifically valuable, particularly for advancing structural biology through room-temperature crystallography. However, while the technical achievements are significant, the manuscript could benefit from more explicit discussions of broader implications for the scientific community, including comparisons to existing methods and potential applications in pharmaceutical research and other fields.

The paper presents an exciting list of innovative approaches. The beamline design, which incorporates a new double-chopper system and allows for 90 μs X-ray pulses at a high repetition rate, is a significant step forward in synchrotron-based structural biology. The ability to switch between sample delivery methods (fixed target and injectors) within an hour is particularly noteworthy, offering flexibility and speed during beamline operation. The introduction of real-time data reduction software is an excellent feature that enhances the efficiency of experiments by identifying crystal hits rapidly. This streamlined approach should improve the overall throughput for users.

Another strength of the work lies in its potential impact on time-resolved studies. The combination of high flux density and shorter exposure times enables high-resolution data collection even from small crystals, opening up new avenues for time-resolved studies of crystal biological reactions. This has the potential to rival femtosecond crystallography at XFELs, which is currently limited by sample consumption and complexity.

The authors also present the possible applications to drug discovery. The successful application of $S\mu X$ to adenosine receptor A2A, co-crystallized with the FDA-approved drug Istradefylline, demonstrates how this technique can aid drug discovery. Room-temperature structures of membrane proteins in complexes with drug molecules are particularly relevant for understanding binding interactions in a near-physiological context.

Although not a showstopper, the manuscript suffers from rare grammatical and stylistic issues. These minor grammatical errors throughout the manuscript detract from the readability. Here are a few examples:

. Abstract, line 8: "ID29 [...] is the first beamline in the world capable of delivering high brilliance microseconds X-ray pulses..." should be corrected to "microsecond X-ray pulses."

. Page 4, line 16: "The collection of diffraction data from hundreds to many thousands of single shot still diffraction patterns..." could be simplified as "The collection of diffraction data from hundreds to thousands of single-shot diffraction patterns..."

. Methods section: Several sentences are long and convoluted. Breaking them into shorter, clearer statements would improve readability.

While the paper is focused on the technical aspects of $S\mu X$, it should provide a clearer explanation of its potential impact on structural biology beyond synchrotron facilities. Although it briefly mentions broad applicability at other synchrotrons (and in

particular other 4th generation synchrotrons), the manuscript could further elaborate on how this technique could be adopted in the pharmaceutical industry, biotechnology, and other areas of research. It is mentioned but this could be further developed.

Similarly, the comparison with existing methods such as XFEL-based crystallography could be expanded. While the differences in photon flux and time resolution are mentioned, further discussion on the practical differences in sample consumption and experimental complexity would be valuable to researchers who may be considering different techniques. While the manuscript discusses how $\text{S}\mu\text{X}$ at ID29 compares with serial femtosecond crystallography at XFELs, the argument would benefit from more quantitative comparisons. For instance, providing detailed data on photon flux, required sample volumes, and achievable time resolutions between the two techniques would strengthen the case for synchrotron-based $\text{S}\mu\text{X}$.

There is little discussion of the limitations of this technique. For example, the manuscript could address further potential challenges, such as radiation damage, particularly for sensitive biological samples at room temperature. Additionally, sample size limitations and difficulties in preparing adequate microcrystals for time-resolved studies might warrant some attention.

The manuscript provides a strong technical foundation, but it could benefit from a more expansive discussion of potential applications. Beyond drug discovery, the authors could speculate on the role of $\text{S}\mu\text{X}$ in studying dynamic processes such as enzyme catalysis, protein folding, and conformational changes in real-time. Furthermore, the manuscript could explore how this technique might facilitate more widespread access to time-resolved crystallography for structural biologists at synchrotron facilities. All these points have been briefly mentioned but leave the reader with a feeling of "I want to know more".

If widely adopted, $\text{S}\mu\text{X}$ could democratize time-resolved structural studies, which have historically been the domain of XFEL facilities. This would enable more laboratories to study dynamic biological processes in real-time, thus accelerating discoveries in structural biology. The manuscript shows promise for enabling time-resolved studies at synchrotrons, but clearer articulation of these broader impacts would help the general scientific community appreciate the significance of this advancement.

I have additional more specific questions concerning some points within the manuscript. I would appreciate it if the authors could answer these points.

Line 19: ESRF or ESRF-EBS?

It is worth clarifying whether the facility should now be consistently referred to as ESRF-EBS. Given that the manuscript highlights the innovations brought about by the Extremely Brilliant Source (EBS) upgrade, using the term ESRF-EBS throughout would be more accurate and reinforce the distinction from the previous generation of the facility.

Line 28: Applicability to 3rd generation synchrotrons

While the manuscript discusses the potential application of the technique at upcoming 4th generation synchrotrons, it would be helpful to address whether similar methods could be adapted or modified for use at powerful 3rd generation synchrotrons. Many facilities still operate with 3rd generation capabilities, and discussing potential modifications or limitations when implementing serial microsecond crystallography ($\text{S}\mu\text{X}$) at these facilities would broaden the relevance of the study.

Line 41: Physiologically relevant conformations

The mention of "identifying physiologically relevant conformations" could benefit from further qualification. The constraints imposed by crystal packing typically prevent the observation of large conformational changes, which are often critical in physiological phenomena. It would be more accurate to note that while $\text{S}\mu\text{X}$ can reveal small conformational changes, the technique may not capture large-scale conformational rearrangements due to the inherent limitations of the crystallographic method in this regard.

Line 43: Vague reference to sample size at XFELs

The phrase "requires a large amount of sample" is vague and lacks specificity. It would be beneficial to provide more context regarding what constitutes a "large" amount relative to the experimental setup, sample type, or comparison to other methods. For instance, quantifying the sample requirements for XFELs in terms of volume or crystal concentration would give readers a clearer understanding of the sample consumption in these experiments. A comparative statement to $\text{S}\mu\text{X}$ or synchrotron-based methods would also be helpful.

Line 52: Time resolution limits at 3rd generation synchrotrons

The statement "limited photon flux density and achievable detector exposure times restrict the achievable time resolution to a few milliseconds" suggests that time resolution at 3rd generation synchrotrons is constrained by current technology. It would be interesting to explore whether advances in flux (such as using multi-color X-rays) and the development of faster detectors could make time-resolved experiments with shorter exposures possible at 3rd generation facilities. Discussing this would provide a forward-looking perspective on the potential for adapting such methodologies in existing synchrotron sources.

Line 74: Inconsistent reporting of flux density increase

There is inconsistency regarding the increase in flux density, with the manuscript mentioning both "5 orders of magnitude" and "3-4 orders of magnitude" in different sections. This discrepancy needs clarification. If the flux increase is context-

dependent (e.g., specific to beamline conditions or sample environments), this should be explicitly mentioned to ensure accuracy and consistency throughout the manuscript.

Line 76: Clarification of signal-to-noise ratios

The term "very high signal-to-noise" is subjective and requires more precise definition. It would be helpful to clarify what constitutes low, medium, high, and very high signal-to-noise ratios within the context of SμX experiments. Providing specific numerical ranges for signal-to-noise ratios would give readers a clearer understanding of the quality of the data being discussed and how this compares to other methods or facilities.

Line 94: Subjective use of the term "Important"

The use of "important" to describe the human integral membrane protein is subjective. It would be more informative to specify why this protein is of interest. For example, mention the biological or pharmacological relevance of the adenosine receptor A2A, particularly its role in drug discovery for diseases such as Parkinson's, to provide a clearer rationale for its inclusion in the study.

Line 107: Reference to a manuscript "in preparation"

The mention of a "manuscript in preparation" is not ideal for a published work as it lacks a definitive reference. The authors should consider either removing this reference or, if the manuscript is close to publication, providing a more concrete citation (e.g., "under review" or "submitted"). If this information is critical to the understanding of the current work, it may be necessary to include more detailed data in the current manuscript instead of deferring to future publications.

Line 112: Clarification of "less monochromatic"

The phrase "less monochromatic" is too vague and should be more specific. It would be helpful to describe the exact degree of polychromaticity or energy bandwidth used in the experiment. For example, mention the percentage of $\Delta E/E$ or energy spread to quantify how "less monochromatic" the beam is compared to traditional monochromatic beamlines, as this detail is crucial for understanding the beamline's performance.

Line 127: Switching time of less than 1 hour

The statement that "switching between fixed-target and injector setups takes less than 1 hour" prompts questions about whether this is a bottleneck for the method. It would be useful to clarify whether this time is a significant limitation for the overall experiment. Additionally, specifying whether sample changes or alignment procedures also require similar timeframes would provide a more comprehensive picture of how time-efficient this system is for users.

Line 146: Correct use of "SPring-8"

The manuscript mentions "SPring-8" in the context of the high-viscosity extruder (HVE). However, it appears that this injector was developed by members of the RIKEN group at SACLA. It would be more accurate to attribute the development correctly to avoid confusion. If the connection to SPring-8 is not directly relevant, the reference should be removed or modified to reflect the proper source.

Line 152: impact of crystal movement on diffraction

The authors mention that the sample moves by less than 0.3 μm during the 90 μs exposure, but there is no discussion of how this movement impacts the diffraction pattern. It would be valuable to analyze whether this small movement affects the quality of the diffraction data, particularly in comparison to more traditional oscillation methods. Providing quantitative analysis or simulation results on this topic would strengthen the manuscript's claims about the accuracy and efficiency of the technique.

Line 153: Photon count and detector values

The phrase "uniform background of maximum 2 photons" may not be meaningful to all structural biologists, as many are more familiar with detector counts. It would be helpful to provide both the photon count and the corresponding detector count (in Analog-Digital Units or ADUs, for example) to make the data more accessible to a broader audience. This would help readers better interpret the impact of the low background noise on the quality of the diffraction data.

Line 186: Similar hits and indexing rates

The statement that "all data presented very similar hits and indexing rates" is questionable. Upon reviewing the reported data, there are noticeable differences in hits and indexing rates, with some datasets showing nearly double the performance of others. This discrepancy should be addressed, either by providing clarification on what is meant by "similar" or by adjusting the language to acknowledge these differences. A more precise statistical analysis or explanation for these variations would be beneficial.

Line 187: B-factors and conformational differences

Instead of broadly stating that there were "no relevant conformational differences with similar overall and site-specific B-factors," it would be more informative to explore the possibility of analyzing B-factor differences between the first part of the exposed data and the last part. This could help in understanding the potential impact of translational diffusion of X-ray photons over the course of the experiment. It might also be worthwhile to clarify whether the assumption that there is no translational diffusion of X-ray photons (due to the diffraction-limited source) is valid in this context, or if further analysis would reveal subtle radiation damage or diffusion effects.

Line 196: Geometrical constraints on resolution

The manuscript mentions that "the resolution was limited by the geometrical constraints at the time of the experiment." It

would be important to clarify if these constraints are likely to be overcome in future experiments. If this is expected to change, the manuscript should discuss how future experiments will address these limitations and what improvements can be anticipated in terms of resolution when readers apply this method in their own research.

Line 198: "Near full occupancy"

The term "near full occupancy" is vague and should be more precisely defined. What level of occupancy is considered "near full," and how might suboptimal occupancy levels affect the interpretation of the structural data? Providing the percentage of ligand occupancy and discussing the potential impact on the quality of the electron density maps and binding interpretations would add clarity.

Line 215: Subjective description of syringes

The phrase "with only 3 syringes" is subjective and could be misinterpreted. The authors should provide a more quantitative description, such as the volume or number of crystals used, to allow readers to understand the sample requirements more accurately. This would make it easier to compare the sample efficiency of the technique to other methods.

Line 260: Calculated dose for experiments

The authors mention that "the calculated dose for the experiments in this work are much higher than the ones typically reported for room temperature measurements," but the exact dose values are not provided. It would be more informative to explicitly state the doses in the manuscript, allowing readers to compare them directly with other studies and evaluate the implications for radiation damage and sample longevity.

Line 266: Future research on sample environments

The statement about "using different sample environments" being part of future research projects is somewhat vague. Either explain in more detail what specific research is planned regarding sample environments or rephrase to focus on the current work. If future research is indeed crucial to this paper's conclusions, providing a brief outline of the intended studies would enhance transparency.

Line 283: Clarifying throughput claims

The statement "one could envisage reaching the throughput of up to 100 ligand-soaked data sets in a 10-hour beamtime with $S_{\mu X}$ " needs clarification. If switching between setups takes up to 1 hour, this could impact the feasibility of achieving such a high throughput. The authors should either revise the math or clarify whether multiple samples can be processed simultaneously, minimizing downtime. A more detailed breakdown of the beamtime and sample handling logistics would help to support this claim.

Line 314: Controlled humidity tent

The mention of using a "controlled humidity tent" raises the question of whether it is compatible with long-term storage and robotic sample mounting. If future setups include sample-holding robots for automation, it is important to assess whether the humidity tent can maintain appropriate conditions for extended periods without affecting sample quality. Clarifying this aspect would give readers a better understanding of the system's flexibility.

Line 630: Acknowledging other injector sources

The manuscript thanks specific individuals for sharing the MPI injector and foil frames but makes no mention of how the other injectors were sourced. Clarifying whether the other devices were commercially available or custom-built for the study would help readers understand the accessibility of the required equipment. If the injectors are available for purchase, providing details about their commercial availability would be helpful in replicating the experiments.

Supplementary Table 2: incorrect R_{work}/R_{free} ratios

In Supplementary Table 2, the R_{work}/R_{free} ratios in the higher resolution shells for datasets like 9FUD and 9FTX appear to be incorrect, with R_{free} values lower than R_{work} . This is unusual since R_{free} is typically expected to be higher than R_{work} . The authors should review and correct these values to ensure the data is accurate and adheres to conventional structural biology standards.

Reviewer #2

(Remarks to the Author)

What are the major claims of the paper?

The major claims are that I29 is a new beamline at a 4th-generation source (ESRF-EBS) and that it measures serial crystallography data on a microsecond timescale. They also claim that measuring data in this way is novel in and of itself and that their data is similar to that of a FEL.

Are the claims novel? If not, please identify the major papers that compromise novelty.

I29 is the first synchrotron beamline at a 4th gen source to measure serial crystallography data on a microsecond timescale. The novelty lies primarily in the 4th generation source and the design of the beamline e.g. the double chopper system. However, this paper does not include the beamline details, which will be published separately. Multiple claims that the beamline is measuring data in 'uncharted and unprecedented ways' is not necessarily the case. Combining a full beamline description with the experimental results would create a more meaningful and novel publication. There is no data supporting microsecond time-scale structural determination.

Will the paper be of interest to others in the field?

Perhaps not in chemistry but to others in the synchrotron field it is interesting.

Are the claims convincing? If not, what further evidence is needed?

The authors often 'oversell' their work and make claims without sufficient evidence. See the more detailed comments below.

Are there other experiments that would strengthen the paper further? How much would they improve it, and how difficult are they likely to be?

1. We need to know how the flux, focal spot size, and pulse width were measured. Plots are required to support this.
2. The beamline diagram lacks detail and does not show the double chopper system mentioned in the text.
3. The authors should provide more evidence that the experimental conditions are unrivalled and leading to a new direction as this statement is currently unsupported.
4. The inclusion of time-resolved data would strengthen the paper significantly.

If the manuscript is unacceptable in its present form, does the study seem sufficiently promising that the authors should be encouraged to consider a resubmission in the future?

The paper is not acceptable in its current form but could be resubmitted if more details can be included. See comments below.

Detailed Comments

Line 14 - ID29, a flagship beamline of the ESRF 4th generation synchrotron, is the first beamline in the world capable of delivering high brilliance microsecond X-ray pulses at high repetition rate for the structure determination of biological macromolecules at room temperature. This is not true as written since XFEL beamlines can also do these experiments. Qualify that the statement refers only to synchrotron beamlines.

Is a new acronym for serial crystallography needed when simply the timescale has changed. Is it not still serial synchrotron crystallography (SSX).

Line 27: revealing the interaction with its inhibitor at the finest details. As a crystallographer, I would expect at the finest details to mean an atomic-resolution electron density map, which is not the case here.

What are the favourable conditions referred to on line 37? And what would the unfavourable ones be?

Line 44. How much is a large amount of sample? Is this not experiment and sample delivery method dependent? Doesn't SX at a synchrotron also require a similar amount of sample to a FEL?

Line 60 Large spectral distribution should be larger spectral dispersion?

Typo costumised – should be customised (line 67), too many commas line 80

Line 73 Wasn't MAX IV the first 4th generation synchrotron? Then Sirius. New era of synchrotron radiation: fourth-generation storage ring | AAPPS Bulletin (springer.com)

When comparing orders of magnitude of flux density (line 73), numbers, what to what? 10^{10} - 10^{15} ?

Line 79 – MAX-IV MicroMAX is open for proposals for serial and time-resolved experiments. SSX is also available at BioMAX

Line 98/99 = complete (how is this defined?) and redundant (a large multiplicity?)

Line 101/102 comparable throughput and efficacy as conventional rotation crystallography – vague/not defined.

Line 112 – double chopper system is not shown in Figure 1.

Line 113 – focal spot size is not shown in Figure 1.

Line 155 what is a reasonably large hit-rate?

Line 196 – resolution was limited by the geometrical constraints. Could the experiments be repeated if the constraint has now been resolved. This would give a better indication of the quality of data that can be achieved at the beamline.

Line 217 as mentioned earlier, at its finer details is too subjective.

Line 203 – 2000 frames was sufficient. How was the limit of ligand density interpretability determined? The ligands' density is quite broken at 2000 frames. Could this be auto-fitted without knowing the ligand's correct position from the full 5000-image dataset?

Line 242 – what does very short pulses mean? This is defined later in the paragraph, but it would be better to state it here. Regardless 'maximising signal-to-noise' is not a feature of the pulse but the jet speed and flux.

Line 247 - The length of time the crystal stays in the beam depends on experiment parameters, for example, jet speed, not pulse width. The pulse width may determine the jet speed.

Line 251/252 – Pulsed beam allows areas exposed to x-rays to be spaced out, also normally happen – see above.

Line 252/253 – seems to play important role in mitigation of radiation damage, ensuring each diffraction image is collected from an unexposed area – also depends on speed.

Line 256/257 – needs a comparison to conventional microfocus beamlines with similar support and mother liquor. Need evidence of no cross-contamination.

Line 258 – flux density achievable at ID29 places it in completely unexplored territory? What does this mean?

Line 258/259 - 9 orders of magnitude lower than FELs and 3 orders higher than 3rd gen synchrotrons and other MX beamlines at next gen light sources? – Need actual values here. Also PETRA is 10^{13} so it is not 3 orders higher.

Lines 260-264 - Why are the calculated doses much higher? Also, if the effect of radiation in RT crystallography is not well-characterised, why not discuss it here?

Line 281 – comparable throughput to conventional MX – the ability to do 100 structures in 10 hours is well below the throughput of conventional MX beamlines, which can easily achieve 25 – 30 samples per hour, and this number will increase dramatically as synchrotrons are upgraded.

Line 284 - One could envisage reaching the throughput of up to 100 ligand-soaked data sets in a 10 hour beamtime with SuX. Based on numbers provided this does not seem feasible, elaborate or remove?

Line 292/293 – it is possible but are you achieving the same spatial resolution and structural quality?

Calculation of dose – line 404 to 422 – need to provide raddose inputs, for example in the supplementary section. Absorbed dose should be in Gy not J/kg

Table 1 seems to be a less informative repeat of the other tables. Why is the data from the extruder the best (resolution, R_{work}, R_{free} etc)? Is it because more images were collected? Or different configuration of the beamline to allow a closer detector distance?

Supplementary table 2

What is the explanation for a high number of hits but the low indexing rate of those hits? What threshold was used to define a hit?

CC* is normally put to 0.7 (or CC1/2 out to 0.3, which is not included). Statistics in table for multiple datasets go past this value, so the resolution cut-off should in fact be greater than 2.05

Plots of these values would help to support this

Average dose per crystal is very high, which is mentioned but not justified. Dose above 100 kGy often considered in literature as 'damaged'

B factors for HEWL si chip and A2AR_istradefylline is very high.

Need beam size, flux and raddose scripts for these experiments.

Why is the detector always set to 150 mm? Should HEWL be able to get resolution down to ~1.7? Why so low?

Supplementary figure 2 – graph intensity should be in photons not ADU – this applies in multiple papers.

Supplementary Table 3 –

The number of collected images is much higher than chip datasets; why?

CC* for HVE_SACLA is 56.98 at the cutoff, why?

The average dose again seems very high

Plots comparing the statistics in addition to the maps would help to support this.

Supplementary table 4 –

IS the CC* listed the overall CC*? Why have they not included the resolution at the cutoff? Completeness is low for A2A-R dataset.

How is this processed? Have you tried using autofit in Phenix suite to see when fitting becomes impossible?

Resolution of each dataset? Fig 4. – do they really have the same resolution cutoff?

Plots of statistics please

Supplementary table 5

What is beam size? Are you assuming top hat or gaussian? Did you change the beam size depending on the crystal size? Because if the beam size for the Injector is smaller, then the average dose per crystal (whole) and (exposed region) should be different

Reviewer #3

(Remarks to the Author)

The manuscript describes the impressive technical work on the ID29 beamline at the ESRF. This instrument is pushing back the frontiers of synchrotron serial crystallography. The manuscript is very well written, cites the literature adequately and contains sufficient technical information to describe the work.

I have a few minor points for correction:

1. Title "... microseconds X-ray pulses ..." should be changed to "... microsecond X-ray pulses ...".
2. Line 388, Data collection parameters; include value of flux at sample.
3. Line 401, "... 5WR880, and 5NM214 respectively." Should be changed to "... 5WR880 and 5NM214, respectively."
4. Lines 683-684: "The absence of "lunes" indicates that no rocking takes place during the acquisition.". Lunes are visible on the left-hand side of Fig2a? Consider re-phrasing or delete the sentence.
5. Lines 181-185 and Fig.3. The difference in observed scattering and B-factors between the 20 micron spacing data and the 10/50 micron data are non-trivial. Is this the result of sample batch variation? The higher scattering and lower B-factors for the 20 micron dataset should be mentioned in the results section of the manuscript.
6. Supplementary Fig3. Panels a,b & c need to be shown at the same magnification as panel d. It is not possible to assess crystal size from panels b and c at such a low magnification.

Apart from a few minor corrections the manuscript, and the work described, are excellent.

Reviewer #4

(Remarks to the Author)

I co-reviewed this manuscript with one of the reviewers who provided the listed reports. This is part of the Communications Chemistry initiative to facilitate training in peer review and to provide appropriate recognition for Early Career Researchers who co-review manuscripts.

Version 1:

Reviewer comments:

Reviewer #1

(Remarks to the Author)

I appreciate the hard work the authors have put into revising the manuscript. They have comprehensively addressed the points raised during the previous review, and the modifications made have improved the clarity and quality of the paper. I am satisfied with the changes and have no further comments. I recommend that the manuscript be accepted for publication in its current form.

Reviewer #2

(Remarks to the Author)

Thank you for addressing our comments in such detail and making changes, including the calculations and the automated ligand fitting calculations. We feel these changes significantly strengthen the manuscript. However, we want to follow up on a few responses.

We acknowledge the comments regarding the restricted resolution limits for some datasets. While we understand that geometrical constraints were the limiting factor, we believe that recollecting the data at a higher resolution—if that is now feasible and not too time-consuming—would better showcase the impressive capabilities of the beamline. It may not be the only data quality indicator, but it is an important metric for most users. Repeating measurements is not necessarily required for publication of this manuscript, but a ~2.05 Å HEWL dataset may not be the best benchmark for your beamline.

The phenix.ligandfit of the GlcNac substrate has a poor RMSD of 2.53Å, which is explained in the text by the rotation along the glucose C2-C5 direction. But could this also be shown in the figure, i.e., a superposition of the ligand as found by phenix.ligandfit vs the correctly fitted ligand? This would show that the ligand fitting was mostly successful despite the large RMSD. Please could you amend the figure to include this?

Thank you for including the CC* plots in your response to us. Could these also be included in the paper? We think this would be informative to readers. Also, please plot these against a linear scale such as 1/d so they can be viewed more clearly. There is indeed no definitive 'correct' threshold for CC*, which is why showing the plots is informative. CC* is not more common for serial crystallography data, it is a measure of the potential quality of the dataset, both CC* and CC ½ are equally valid. From your CC* plots, some of your data seem to have a low correlation in the low-resolution shells. Please

could you also include in the manuscript the peakogram plot from crystfel to show you have not overloaded your detector; this is a common reason for this pathology. If it turns out you do have overloads, perhaps this would be another good reason to recollect the data.

You frequently state that data collected on ID29 is comparable to data collected at an XFEL, however, it is perhaps equally informative to see how these data compare to data collected at 3rd generation sources. For example, <https://pubs.aip.org/aip/rsi/article/90/8/085110/360302/The-serial-millisecond-crystallography-instrument>, was collected at the Australian Synchrotron. In this paper they collected Lysozyme data to 1.83 Å. This is by no means an exhaustive search. Still, if the goal of this paper is to demonstrate the benefits of ID29 over previous beamlines, then this may be another good reason to recollect your data, given it takes ~20 minutes to collect a full dataset (Table 1). If you want to compare your data to SFX data collected at XFELs, please include a genuine comparison of example data from an XFEL in the manuscript to support this statement. As we have stated, we would also advise making a similar comparison with SSX data from 3rd generation sources. There will obviously be differences in sample delivery method, etc, but these can be discussed in the manuscript. This will give the reader a better idea of where this beamline sits compared to its competitors.

We would like to try again to clearly state that the time a sample is exposed to the beam is dictated by a combination of the delivery method and the source. The manuscript currently implies that the exposure time of ID29, means that the crystal is continuously in the beam during an exposure, and that this could not occur at other sources. This is simply not true for all delivery methods. For example, jets can be run slower so crystals spend more time in the beam, and fixed targets can literally stop moving such that the crystal can be immobilized in the beam for as long an exposure time as required. Please rewrite this point, specifying that with ID29 jets can be run faster to achieve this goal or simply delete this point.

While we can't account for this paper you mentioned in your rebuttal, (Zhao, et al 2019. "A Guide to Sample Delivery Systems for Serial Crystallography." The FEBS Journal 286 (22): 4402–17. <https://doi.org/10.1111/febs.15099>), but can you state reasons in the manuscript as to why, with the same sample delivery method, your consumption is less than an XFEL? Assuming you optimized your rate of sample delivery to your source, there should be no difference in sample consumption at an XFEL, a 4th generation source or a 3rd generation source. The only thing that changes is the rate at which you will acquire data and, perhaps, the stability of your jet. It is true, when you introduce a laser pump pulse into an experiment, this will change the required jet speed. If this is the point you wish to make, you need to mention the trade-offs between pump pulse and jet speed.

Please plot radial integrals in photon counts or ADUs consistently throughout your manuscript. Preferably in photon counts as this is more recognizable across more detector types.

Reviewer #3

(Remarks to the Author)

The revised manuscript has addressed this reviewers corrections and is suitable for publication.

Reviewer #4

(Remarks to the Author)

I co-reviewed this manuscript with one of the reviewers who provided the listed reports. This is part of the Communications Chemistry initiative to facilitate training in peer review and to provide appropriate recognition for Early Career Researchers who co-review manuscripts.

Version 2:

Reviewer comments:

Reviewer #2

(Remarks to the Author)

Thank you for revising the manuscript for a second time. I think we can agree to disagree on some points, but I would be happy to see the paper published as it stands.

Dear Dr Guo,

We would like to express our sincere gratitude to you and the four reviewers for your thorough review and insightful comments on our manuscript titled "Advancing macromolecular structure determination with microseconds X-ray pulses at a 4th generation synchrotron" (COMMSCHEM-24-0459-T). We have carefully considered all the feedback and have made the necessary revisions accordingly. We enclose an amended version of the manuscript, where all changes are highlighted. Below, we provide a detailed response to each of the comments raised by the reviewers, as well as a summary of the changes made in the manuscript.

Reviewers' comments:

Reviewer #1 (Remarks to the Author):

The manuscript from de Sanctis et al. titled "Advancing macromolecular structure determination with microseconds X-ray pulses at a 4th generation synchrotron" presents the development and application of serial microsecond crystallography (S μ X) using high-brilliance, high-repetition-rate X-ray pulses at the newly constructed ID29 beamline at the ESRF-EBS. It introduces innovations in beamline design, sample delivery methods, and real-time data acquisition techniques, with applications to model systems and integral membrane proteins.

The work is scientifically valuable, particularly for advancing structural biology through room-temperature crystallography. However, while the technical achievements are significant, the manuscript could benefit from more explicit discussions of broader implications for the scientific community, including comparisons to existing methods and potential applications in pharmaceutical research and other fields.

The paper presents an exciting list of innovative approaches. The beamline design, which incorporates a new double-chopper system and allows for 90 μ s X-ray pulses at a high repetition rate, is a significant step forward in synchrotron-based structural biology. The ability to switch between sample delivery methods (fixed target and injectors) within an hour is particularly noteworthy, offering flexibility and speed during beamline operation. The introduction of real-time data reduction software is an excellent feature that enhances the efficiency of experiments by identifying crystal hits rapidly. This streamlined approach should improve the overall throughput for users.

Another strength of the work lies in its potential impact on time-resolved studies. The combination of high flux density and shorter exposure times enables high-resolution data collection even from small crystals, opening up new avenues for time-resolved studies of crystal biological reactions. This has the potential to rival femtosecond crystallography at XFELs, which is currently limited by sample consumption and complexity.

The authors also present the possible applications to drug discovery. The successful application of S μ X to adenosine receptor A2A, co-crystallized with the FDA-approved drug Istradefylline, demonstrates how this technique can aid drug discovery. Room-temperature structures of membrane proteins in complexes with drug molecules are particularly relevant for understanding binding interactions in a near-physiological context.

R: We thank the reviewer for the positive feedback and consideration of the impact of the developments and results presented in the manuscript.

Although not a showstopper, the manuscript suffers from rare grammatical and stylistic issues. These minor grammatical errors throughout the manuscript detract from the readability. Here are a few examples:

- . Abstract, line 8: "ID29 [...] is the first beamline in the world capable of delivering high brilliance microseconds X-ray pulses..." should be corrected to "microsecond X-ray pulses."
- . Page 4, line 16: "The collection of diffraction data from hundreds to many thousands of single shot still diffraction patterns..." could be simplified as "The collection of diffraction data from hundreds to thousands of single-shot diffraction patterns..."
- . Methods section: Several sentences are long and convoluted. Breaking them into shorter, clearer statements would improve readability.

R: Thank you for pointing this out, we have amended the text following your recommendations.

While the paper is focused on the technical aspects of S μ X, it should provide a clearer explanation of its potential impact on structural biology beyond synchrotron facilities. Although it briefly mentions broad applicability at other synchrotrons (and in particular other 4th generation synchrotrons), the manuscript could further elaborate on how this technique could be adopted in the pharmaceutical industry, biotechnology, and other areas of research. It is mentioned but this could be further developed.

R: Thanks for the reviewer's constructive comment, we have modified the sentence to highlight the importance that this approach could represent for proprietary research.

Similarly, the comparison with existing methods such as XFEL-based crystallography could be expanded. While the differences in photon flux and time resolution are mentioned, further discussion on the practical differences in sample consumption and experimental complexity would be valuable to researchers who may be considering different techniques. While the manuscript discusses how S μ X at ID29 compares with serial femtosecond crystallography at XFELs, the argument would benefit from more quantitative comparisons. For instance, providing detailed data on photon flux, required sample volumes, and achievable time resolutions between the two techniques would strengthen the case for synchrotron-based S μ X.

R: We think we have addressed this point in the Discussion, where characteristics of the achievable time resolutions are compared. Regarding the point on sample consumption, please see our reply to a similar comment below.

There is little discussion of the limitations of this technique. For example, the manuscript could address further potential challenges, such as radiation damage, particularly for sensitive biological samples at room temperature. Additionally, sample size limitations and difficulties in preparing adequate microcrystals for time-resolved studies might warrant some attention.

R: This is a very good point brought up by the reviewer. We have actually mentioned the radiation damage aspect for this technique in the discussion section. As the radiation damage coupled with microsecond pulses is an unexplored area of research, we believe a dedicated analysis would be necessary to fully comprehend it. This is beyond the scope of our current work. Nevertheless, we have provided preliminary results in this manuscript, showing radiation damage can be outrun for real-life samples. Additionally, challenges due to microcrystal size and quantity are already being mentioned in the literature, which are cited in relevant texts of the manuscript. However, we have shown the example of A2A receptor, where we had a limited amount of samples, and yet were able to determine the ligand binding.

The manuscript provides a strong technical foundation, but it could benefit from a more expansive discussion of potential applications. Beyond drug discovery, the authors could speculate on the role of S μ X in studying dynamic processes such as enzyme catalysis, protein folding, and conformational changes in real-time. Furthermore, the manuscript could explore how this technique might facilitate more widespread access to time-resolved crystallography for structural biologists at synchrotron facilities. All these points have been briefly mentioned but leave the reader with a feeling of "I want to know more".

R: We appreciate the constructive remark. We have further developed the discussion to include the points mentioned by the reviewer.

If widely adopted, S μ X could democratize time-resolved structural studies, which have historically been the domain of XFEL facilities. This would enable more laboratories to study dynamic biological processes in real-time, thus accelerating discoveries in structural biology. The manuscript shows promise for enabling time-resolved studies at synchrotrons, but clearer articulation of these broader impacts would help the general scientific community appreciate the significance of this advancement.

R: We thank the reviewer for seeing the impact that such a research field can have on the structural biology community. We agree with the statement and we have developed the Discussion section to highlight this point.

I have additional more specific questions concerning some points within the manuscript. I would appreciate it if the authors could answer these points.

Line 19: ESRF or ESRF-EBS?

It is worth clarifying whether the facility should now be consistently referred to as ESRF-EBS. Given that the manuscript highlights the innovations brought about by the Extremely Brilliant

Source (EBS) upgrade, using the term ESRF-EBS throughout would be more accurate and reinforce the distinction from the previous generation of the facility.

R: The facility name is still ESRF, the term ESRF-EBS refers to the upgrade and to the new accelerator machine. We use ESRF-EBS when we are referring to the new machine to differentiate from its predecessor and ESRF when referring to the facility.

Line 28: Applicability to 3rd generation synchrotrons

While the manuscript discusses the potential application of the technique at upcoming 4th generation synchrotrons, it would be helpful to address whether similar methods could be adapted or modified for use at powerful 3rd generation synchrotrons. Many facilities still operate with 3rd generation capabilities, and discussing potential modifications or limitations when implementing serial microsecond crystallography (S μ X) at these facilities would broaden the relevance of the study.

R: This is a pertinent observation and in fact a similar concept could possibly be applied to 3rd generation sources, although with longer exposure because of the lower flux. We have added a sentence in the discussion to highlight this.

Line 41: Physiologically relevant conformations

The mention of "identifying physiologically relevant conformations" could benefit from further qualification. The constraints imposed by crystal packing typically prevent the observation of large conformational changes, which are often critical in physiological phenomena. It would be more accurate to note that while S μ X can reveal small conformational changes, the technique may not capture large-scale conformational rearrangements due to the inherent limitations of the crystallographic method in this regard.

R: We completely agree with the reviewer's comment. By physiologically relevant we meant (as the cited references) free from cryo-protectant and cryogenic artifacts. We have however amended the sentence to specify that this is limited to reactions that are compatible with the crystalline form (line 44)

Line 43: Vague reference to sample size at XFELs

The phrase "requires a large amount of sample" is vague and lacks specificity. It would be beneficial to provide more context regarding what constitutes a "large" amount relative to the experimental setup, sample type, or comparison to other methods. For instance, quantifying the sample requirements for XFELs in terms of volume or crystal concentration would give readers a clearer understanding of the sample consumption in these experiments. A comparative statement to S μ X or synchrotron-based methods would also be helpful.

R: The sample consumption for XFEL experiments is a known concern and different developments are ongoing to address it. However sample delivery is constantly evolving and there is a high variability across different methods and samples, for this reason it is hard to give absolute values. We have nevertheless added a citation of a review

<https://febs.onlinelibrary.wiley.com/doi/epdf/10.1111/febs.15099>) that compares different sample delivery methods both at XFELs and synchrotrons (lines 44-47).

Line 52: Time resolution limits at 3rd generation synchrotrons

The statement "limited photon flux density and achievable detector exposure times restrict the achievable time resolution to a few milliseconds" suggests that time resolution at 3rd generation synchrotrons is constrained by current technology. It would be interesting to explore whether advances in flux (such as using multi-color X-rays) and the development of faster detectors could make time-resolved experiments with shorter exposures possible at 3rd generation facilities. Discussing this would provide a forward-looking perspective on the potential for adapting such methodologies in existing synchrotron sources.

R: We have added a sentence in the Discussion to consider this point (lines 254-255).

Line 74: Inconsistent reporting of flux density increase

There is inconsistency regarding the increase in flux density, with the manuscript mentioning both "5 orders of magnitude" and "3-4 orders of magnitude" in different sections. This discrepancy needs clarification. If the flux increase is context-dependent (e.g., specific to beamline conditions or sample environments), this should be explicitly mentioned to ensure accuracy and consistency throughout the manuscript.

R: Indeed these numbers compare different beamlines and setup. The "5 order of magnitudes" referred to the original ID29 MAD beamline, we have removed this comparison and now we refer to 2 - 4 orders of magnitude in comparison with microfocus beamlines at both 3rd and 4th generation synchrotrons (line 76 and 277)

Line 76: Clarification of signal-to-noise ratios

The term "very high signal-to-noise" is subjective and requires more precise definition. It would be helpful to clarify what constitutes low, medium, high, and very high signal-to-noise ratios within the context of SpX experiments. Providing specific numerical ranges for signal-to-noise ratios would give readers a clearer understanding of the quality of the data being discussed and how this compares to other methods or facilities.

R: We agree with the reviewer's comment, and obviously the signal-to-noise ratios depend on many factors. Since this sentence is part of the Introduction and we are referring to individual intensities it would be inappropriate to give a specific number. We have hence changed this sentence and refer to the Results session for a quantitative analysis (lines 157-159).

Line 94: Subjective use of the term "Important"

The use of "important" to describe the human integral membrane protein is subjective. It would be more informative to specify why this protein is of interest. For example, mention the biological or pharmacological relevance of the adenosine receptor A2A, particularly its role in drug discovery for diseases such as Parkinson's, to provide a clearer rationale for its inclusion in the study.

R: Point taken, we have developed the paragraph to highlight the importance of Istradefylline and added a reference (lines 98-102 and lines 217-219).

Line 107: Reference to a manuscript "in preparation"

The mention of a "manuscript in preparation" is not ideal for a published work as it lacks a definitive reference. The authors should consider either removing this reference or, if the manuscript is close to publication, providing a more concrete citation (e.g., "under review" or "submitted"). If this information is critical to the understanding of the current work, it may be necessary to include more detailed data in the current manuscript instead of deferring to future publications.

R: We have removed the sentence.

Line 112: Clarification of "less monochromatic"

The phrase "less monochromatic" is too vague and should be more specific. It would be helpful to describe the exact degree of polychromaticity or energy bandwidth used in the experiment. For example, mention the percentage of $\Delta E/E$ or energy spread to quantify how "less monochromatic" the beam is compared to traditional monochromatic beamlines, as this detail is crucial for understanding the beamline's performance.

R: We are not sure we understood the comment, as in the same sentence we specify that the bandwidth is 1 % $\Delta E/E$ (line 79).

Line 127: Switching time of less than 1 hour

The statement that "switching between fixed-target and injector setups takes less than 1 hour" prompts questions about whether this is a bottleneck for the method. It would be useful to clarify whether this time is a significant limitation for the overall experiment. Additionally, specifying whether sample changes or alignment procedures also require similar timeframes would provide a more comprehensive picture of how time-efficient this system is for users.

R: We believe the reviewer misinterpreted the sentence, the switching time refers to change the setup from fixed-target experiments to injector-based experiments. The time needed to change samples mounted on a fixed-target support (Si-chip or foil) is only few seconds.

Line 146: Correct use of "SPring-8"

The manuscript mentions "SPring-8" in the context of the high-viscosity extruder (HVE). However, it appears that this injector was developed by members of the RIKEN group at SACLA. It would be more accurate to attribute the development correctly to avoid confusion. If the connection to Spring-8 is not directly relevant, the reference should be removed or modified to reflect the proper source.

R: Thanks to the reviewer for clarification. We have updated the affiliation to "RIKEN Spring-8 Center (SACLA)" as reported in the reference (lines 151-152).

Line 152: impact of crystal movement on diffraction

The authors mention that the sample moves by less than 0.3 μm during the 90 μs exposure, but there is no discussion of how this movement impacts the diffraction pattern. It would be valuable to analyze whether this small movement affects the quality of the diffraction data, particularly in comparison to more traditional oscillation methods. Providing quantitative analysis or simulation results on this topic would strengthen the manuscript's claims about the accuracy and efficiency of the technique.

R: What we meant is that during the exposure the crystal is constantly in the beam and this maximizes the signal to noise ratio. If the crystal was moving by more than its size or beam size during the exposure, the detector would acquire for a time longer than the time the crystal could diffract, increasing the background and thus reducing the signal to noise. Regarding the impact of the movement in the diffraction, this is no concern as the movement is performed at constant speed. This was already the case for continuous helical scan (see <https://doi.org/10.1107/S0909049512009715> <https://doi.org/10.1107/S0907444910028192> <https://doi.org/10.1107/S2052252513033939> to name some). Supplementary Table 1 reports the speed and the displacement of the crystals in the different data acquisition schemes we presented.

Line 153: Photon count and detector values

The phrase "uniform background of maximum 2 photons" may not be meaningful to all structural biologists, as many are more familiar with detector counts. It would be helpful to provide both the photon count and the corresponding detector count (in Analog-Digital Units or ADUs, for example) to make the data more accessible to a broader audience. This would help readers better interpret the impact of the low background noise on the quality of the diffraction data.

R: We believe that since the advent of photon counting detectors the community got familiar with the use of the number of photons. Moreover we refer to Figure 2, the legend of which specifies that 1 photon corresponds to 478 ADU (line 712).

Line 186: Similar hits and indexing rates

The statement that "all data presented very similar hits and indexing rates" is questionable. Upon reviewing the reported data, there are noticeable differences in hits and indexing rates, with some datasets showing nearly double the performance of others. This discrepancy should be addressed, either by providing clarification on what is meant by "similar" or by adjusting the language to acknowledge these differences. A more precise statistical analysis or explanation for these variations would be beneficial.

R: in this section we are referring to the Proteinase K foils data collection which presents an indexing rate between 3 and 6 % which we considered very similar. Thanks to the reviewer's suggestion, we have explicitly included the hits/indexed rates in the main text (line 192).

Line 187: B-factors and conformational differences

Instead of broadly stating that there were "no relevant conformational differences with similar overall and site-specific B-factors," it would be more informative to explore the possibility of analyzing B-factor differences between the first part of the exposed data and the last part. This could help in understanding the potential impact of translational diffusion of X-ray photons over the course of the experiment. It might also be worthwhile to clarify whether the assumption that there is no translational diffusion of X-ray photons (due to the diffraction-limited source) is valid in this context, or if further analysis would reveal subtle radiation damage or diffusion effects.

R: Although this is an interesting comment from the reviewer, we still may not be able to entirely comprehend what the reviewer means by "translational diffusion of X-ray photons" and how this might be related to the diffraction-limited source. However, we would like to note that we have analyzed the intensity of the diffraction and monitor radiation sensitive sites at different spacings (as in Fig.3) and this shows no difference for the 10 to 50 μm spacing, ensuring no damage propagation. Please note that the mean photoelectron path length at the energy of the experiments (i.e., 11.56 keV) is below 1 μm (Please see Nave, Colin, and Mark A. Hill. 2005. "Will Reduced Radiation Damage Occur with Very Small Crystals?" *Journal of Synchrotron Radiation* 12 (Pt 3): 299–303. <https://doi.org/10.1107/S0909049505003274> and Dickerson, Joshua L., and Elspeth F. Garman. 2021. "Doses for Experiments with Microbeams and Microcrystals: Monte Carlo Simulations in RADDOS-3D." *Protein Science: A Publication of the Protein Society* 30 (1): 8–19. <https://doi.org/10.1002/pro.3922>). This clearly explains our observation in Fig 3. These references are cited in the revised manuscript for further clarity.

Line 196: Geometrical constraints on resolution

The manuscript mentions that "the resolution was limited by the geometrical constraints at the time of the experiment." It would be important to clarify if these constraints are likely to be overcome in future experiments. If this is expected to change, the manuscript should discuss how future experiments will address these limitations and what improvements can be anticipated in terms of resolution when readers apply this method in their own research.

R: The resolution was limited as the sample-to-detector distance was fixed to 150 mm. Eventually the detector was installed on a new motorized support to achieve the minimal distance of 100 mm for a resolution of 1.6 \AA . The text has been amended to clarify this point (lines 203-205).

Line 198: "Near full occupancy"

The term "near full occupancy" is vague and should be more precisely defined. What level of occupancy is considered "near full," and how might suboptimal occupancy levels affect the interpretation of the structural data? Providing the percentage of ligand occupancy and discussing the potential impact on the quality of the electron density maps and binding interpretations would add clarity.

R: Good point. We have amended the text to specify the exact value obtained from occupancy refinement with phenix.refine (line 207).

Line 215: Subjective description of syringes

The phrase "with only 3 syringes" is subjective and could be misinterpreted. The authors should provide a more quantitative description, such as the volume or number of crystals used, to allow readers to understand the sample requirements more accurately. This would make it easier to compare the sample efficiency of the technique to other methods.

R: We understand, but in the same sentence we specified that this corresponded to 15 ul of LCP. However, we have rephrased this point to "With only 15 μ L of LCP, pooled from three syringes" (lines 227-228).

Line 260: Calculated dose for experiments

The authors mention that "the calculated dose for the experiments in this work are much higher than the ones typically reported for room temperature measurements," but the exact dose values are not provided. It would be more informative to explicitly state the doses in the manuscript, allowing readers to compare them directly with other studies and evaluate the implications for radiation damage and sample longevity.

R: In fact, the dose for each dataset is provided in Supplementary Table 5. As there are many datasets presented in the manuscript, we preferred not to list the dose values in the main text. Also in the main text, we have referred them explicitly to Supplementary Table 2-3 and Supplementary Table 5 at this point of the Discussion.

Line 266: Future research on sample environments

The statement about "using different sample environments" being part of future research projects is somewhat vague. Either explain in more detail what specific research is planned regarding sample environments or rephrase to focus on the current work. If future research is indeed crucial to this paper's conclusions, providing a brief outline of the intended studies would enhance transparency.

R: We have modified the sentence to highlight that the characterisation of radiation damage in beamlines like ID29 is an important field of research to design future experiments and design of similar instruments (lines 282-284).

Line 283: Clarifying throughput claims

The statement "one could envisage reaching the throughput of up to 100 ligand-soaked data sets in a 10-hour beamtime with SpX" needs clarification. If switching between setups takes up to 1 hour, this could impact the feasibility of achieving such a high throughput. The authors should either revise the math or clarify whether multiple samples can be processed simultaneously, minimizing downtime. A more detailed breakdown of the beamtime and sample handling logistics would help to support this claim.

R: We highly appreciate the reviewer's comment on high throughput. We should clarify that the switching time refers to the whole change of setup between the fixed target configuration and the injector configuration, not to exchange fixed target supports. The time to exchange the fixed target supports is well under a minute. Once automated through sample-changing robots, this is going to be as fast as an automounter on a cryo crystallography beamline.

Line 314: Controlled humidity tent

The mention of using a "controlled humidity tent" raises the question of whether it is compatible with long-term storage and robotic sample mounting. If future setups include sample-holding robots for automation, it is important to assess whether the humidity tent can maintain appropriate conditions for extended periods without affecting sample quality. Clarifying this aspect would give readers a better understanding of the system's flexibility.

R: The chips were loaded in a controlled humidity tent because blotting reduces the amount of surrounding liquid and dehydration might impair the crystal diffraction quality. This is a normal procedure when loading this kind of support (see references 38 and 40 for Si chips and HARE chips reported in the manuscript). The development of a robot mounting is still in an early stage, but we have clarified that the chips would have to be stored in a humidity-controlled storage chamber.

Line 630: Acknowledging other injector sources

The manuscript thanks specific individuals for sharing the MPI injector and foil frames but makes no mention of how the other injectors were sourced. Clarifying whether the other devices were commercially available or custom-built for the study would help readers understand the accessibility of the required equipment. If the injectors are available for purchase, providing details about their commercial availability would be helpful in replicating the experiments.

R: We thank the reviewer for the comment. In fact the MPI injector and frames were kindly provided, while the other injectors were purchased. We have amended the text in the Materials and Methods section to clarify this (line 295).

Supplementary Table 2: incorrect Rwork/Rfree ratios

In Supplementary Table 2, the Rwork/Rfree ratios in the higher resolution shells for datasets like 9FUD and 9FTX appear to be incorrect, with Rfree values lower than Rwork. This is unusual since Rfree is typically expected to be higher than Rwork. The authors should review and correct these values to ensure the data is accurate and adheres to conventional structural biology standards.

R: We are thankful for pointing out the errors. Indeed we reported the wrong values. We apologize for the mistake. The Rwork/Rfree in the higher resolution have been corrected for these two structures in Supplementary Tables.

Reviewer #2 (Remarks to the Author):

What are the major claims of the paper?

The major claims are that I29 is a new beamline at a 4th-generation source (ESRF-EBS) and that it measures serial crystallography data on a microsecond timescale. They also claim that measuring data in this way is novel in and of itself and that their data is similar to that of a FEL.

Are the claims novel? If not, please identify the major papers that compromise novelty.

I29 is the first synchrotron beamline at a 4th gen source to measure serial crystallography data on a microsecond timescale. The novelty lies primarily in the 4th generation source and the design of the beamline e.g. the double chopper system. However, this paper does not include the beamline details, which will be published separately. Multiple claims that the beamline is measuring data in 'uncharted and unprecedented ways' is not necessarily the case. Combining a full beamline description with the experimental results would create a more meaningful and novel publication. There is no data supporting microsecond time-scale structural determination.

Will the paper be of interest to others in the field?

Perhaps not in chemistry but to others in the synchrotron field it is interesting.

Are the claims convincing? If not, what further evidence is needed?

The authors often 'oversell' their work and make claims without sufficient evidence. See the more detailed comments below.

Are there other experiments that would strengthen the paper further? How much would they improve it, and how difficult are they likely to be?

1. We need to know how the flux, focal spot size, and pulse width were measured. Plots are required to support this.
2. The beamline diagram lacks detail and does not show the double chopper system mentioned in the text.
3. The authors should provide more evidence that the experimental conditions are unrivalled and leading to a new direction as this statement is currently unsupported.
4. The inclusion of time-resolved data would strengthen the paper significantly.

R: The flux was measured using a high voltage ionization chamber as described in Sato, et al 1998. "The Behavior of Ionization Chambers and the Criterion of High Applied Voltage under the High Current Storage Ring Operation,"

http://www.spring8.or.jp/pdf/en/ann_rep/97/P225-227.pdf. The ionization chamber was filled with N at 0.8 bar with an applied voltage of 4 kV. The current was measured with a Keithley 2picoammeter. The focal spot size using a knife-edge scan and the pulse width was obtained by the current of a polarised diode placed behind a carbon foil, converted to voltage and measured with an oscilloscope.

The beamline diagram contains only the heat-load chopper that was used to generate the 90 μ s pulse and therefore for clarity we decided to only display one chopper. We have amended the text to clarify this point

Regarding the unrivaled experimental conditions, we are not sure how we should address this comment, as ID29 is the only beamline at the moment capable of producing microsecond pulses with a flux $> 10^{15}$ ph/s in a 1 % bandwidth. Similar beamlines are starting their commissioning or are in the design phase

(<https://journals.iucr.org/s/issues/2018/03/00/xe5028/index.html>,
<https://www.maxiv.lu.se/beamlines-accelerators/beamlines/micromax/>,
<https://lnls.cnpem.br/facilities/manaca-en/>

<https://www.cells.es/en/instruments/beamlines/bl06-xaira> to name some) and this is why we claim this will lead to new opportunities for serial crystallography at new sources.

The goal of this paper was to highlight the technical advances of ID29 and showcase the possibility to collect data with a pulsed beam at 4th generation synchrotron.

If the manuscript is unacceptable in its present form, does the study seem sufficiently promising that the authors should be encouraged to consider a resubmission in the future?

The paper is not acceptable in its current form but could be resubmitted if more details can be included. See comments below.

Detailed Comments

Line 14 - ID29, a flagship beamline of the ESRF 4th generation synchrotron, is the first beamline in the world capable of delivering high brilliance microsecond X-ray pulses at high repetition rate for the structure determination of biological macromolecules at room temperature. This is not true as written since XFEL beamlines can also do these experiments. Qualify that the statement refers only to synchrotron beamlines.

R: The sentence is referring to synchrotrons, for sake of clarity we have amended the text to reiterate this (line 19).

Is a new acronym for serial crystallography needed when simply the timescale has changed. Is it not still serial synchrotron crystallography (SSX).

R: Serial synchrotron crystallography is a very broad term that includes different ways of performing serial data collection at synchrotrons. One of which is Serial Millisecond Crystallography (Weinert et al. 2017. "Serial Millisecond Crystallography for Routine Room-Temperature Structure Determination at Synchrotrons." *Nature Communications* 8 (1): 542. <https://doi.org/10.1038/s41467-017-00630-4>.) We believe that the specificity of the experiments that are performed at ID29, producing true microsecond pulses, are well identified by the new acronym.

Line 27: revealing the interaction with its inhibitor at the finest details. As a crystallographer, I would expect at the finest details to mean an atomic-resolution electron density map, which is not the case here.

R: We rephrase this sentence (lines 27-28).

What are the favourable conditions referred to on line 37? And what would the unfavourable ones be?

R: With favourable conditions we meant that the crystal slurry is well concentrated, that the X-ray pulses are hitting the samples and that the crystals have a good diffracting power.

Line 44. How much is a large amount of sample? Is this not experiment and sample delivery method dependent? Doesn't SX at a synchrotron also require a similar amount of sample to a FEL?

R: Usually serial crystallography at synchrotron requires less sample than serial crystallography at XFELs (see Zhao, et al 2019. "A Guide to Sample Delivery Systems for Serial Crystallography." *The FEBS Journal* 286 (22): 4402–17. <https://doi.org/10.1111/febs.15099>). In our results, we show that at ID29 the amount of required sample is even lower, as demonstrated in the case of A_{2A}R in complex with Istradefylline, which could be fully determined with only 15 µl of lipidic cubic phase bolus.

Line 60 Large spectral distribution should be larger spectral dispersion?

R: We prefer to use the term distribution, in fact spectral dispersion describes the separation of light in different wavelengths.

Typo costumised – should be customised (line 67), too many commas line 80

R: We have amended the text.

Line 73 Wasn't MAX IV the first 4th generation synchrotron? Then Sirius. New era of synchrotron radiation: fourth-generation storage ring | AAPPS Bulletin (springer.com)

R: This is correct, in fact ESRF-EBS is the first high energy 4th generation synchrotron, based on the Hybrid Multi Bend Achromat (HMBA) lattice. In the paper that the reviewer is citing (<https://doi.org/10.1007/s43673-021-00021-4>) is reported: "The commissioning of the ESRF-EBS had been extremely successful and on August 25, 2020, user service mode operations started as planned. In addition to MAX-IV and SIRIUS, the first 6-GeV dream machine had come to life and delivered ultra-small X-ray beams". For further clarification, we explicitly report the energy of the storage ring (6 GeV) in the main text (line 74).

When comparing orders of magnitude of flux density (line 73), numbers, what to what? 10¹⁰-10¹⁵?

R: We have amended the text to clarify this point, since in the original version we were comparing also with the old ID29 beamline. This has been removed and we are now comparing ID29 to microfocus beamlines with similar focusing capabilities (line 77).

Line 79 – MAX-IV MicroMAX is open for proposals for serial and time-resolved experiments. SSX is also available at BioMAX

R: At the time of writing MicroMAX was not yet open for proposals. To the best of our knowledge, BioMAX is performing SSX experiments upon special user requests as part of their research activity instead of a dedicated service, similarly to other microfocus beamlines (such as ID23-2 at ESRF and I24 at Diamond Light Source) but does not present a large bandwidth beam and microsecond pulses. We have amended the text to highlight the 4th generation sources that are in operation (line2 89-90).

Line 98/99 = complete (how is this defined?) and redundant (a large multiplicity?)

R: As this sentence is still part of the Introduction we did not enter into detailed values which are instead reported in the Results section and Supplementary tables. We have changed redundant with high multiplicity (line 105).

Line 101/102 comparable throughput and efficacy as conventional rotation crystallography – vague/not defined.

R: Rotational cryogenic crystallography at synchrotrons is now conventional, over the years it has become possible to collect data quicker and quicker because of the introduction of fast-photon counting detectors, shutterless collection, robotics, automation etc. Here, we envisage that SpX will transform serial crystallography into a more accessible approach with comparable throughputs. This is further explained and developed in the Discussion section based on our results (lines 301-302).

Line 112 – double chopper system is not shown in Figure 1.

R: The second chopper was in commissioning at the time of the experiments (as reported in the Results) and not used for the data acquisition reported in the manuscript, for this reason it was not included in Figure 1. We have amended the text to clarify this (line 102).

Line 113 – focal spot size is not shown in Figure 1.

R: We are not sure we understand the comment, since the spot size is around $4 \times 2 \mu\text{m}^2$ it would not be visible in Figure 1. However, we have added the X-ray beam size in the Figure 1 legend.

Line 155 what is a reasonably large hit-rate?

R: Of course the hit rate depends on the sample delivery methods. Generally a hit rate of 10% is considered efficient. Therefore the hit rate referred to this point is well above this threshold. We also have added the percentage of the hit-rate based on the values reported in Supplementary Table 3.

Line 196 – resolution was limited by the geometrical constraints. Could the experiments be repeated if the constraint has now been resolved. This would give a better indication of the quality of data that can be achieved at the beamline.

R: Although there is a limitation in the achievable resolution, this is not an indicator of data quality, which is rather reflected in the quality of the map (and in our case in the quality of the electron density of the ligand). The geometrical constraints limited the sample-to-detector distance to 150 mm.

Line 217 as mentioned earlier, at its finer details is too subjective.

R: We have removed this statement.

Line 203 – 2000 frames was sufficient. How was the limit of ligand density interpretability determined? The ligands' density is quite broken at 2000 frames. Could this be auto-fitted without knowing the ligand's correct position from the full 5000-image dataset?

R: Following the reviewer's suggestion, phenix.ligandfit was used to successfully place the ligands in the 2000 frames datasets for both Lysozyme:GlcNac and A_{2A}R:Istradefyllin. This is included in the main text together with the rms deviation of the positioning of the ligand from the final one (lines 211-216 and lines 234-236).

Line 242 – what does very short pulses mean? This is defined later in the paragraph, but it would be better to state it here. Regardless, 'maximising signal-to-noise' is not a feature of the pulse but the jet speed and flux.

R: In the paragraph just before we have mentioned pulses of "tens of microseconds" and we have now specified again that in the reported experiments the pulse is of 90 μs. Regarding the "signal-to-noise" please see our explanation on the next point (line 260).

Line 247 - The length of time the crystal stays in the beam depends on experiment parameters, for example, jet speed, not pulse width. The pulse width may determine the jet speed.

R: We are not sure we fully understand the reviewer comment. We agree that the time the crystal stays in the beam depends on the jet speed (and vertical beam size), but here we are not considering the time the crystal stays in the beam relative to the frame exposure time. Since the time the crystal stays in the beam is the same, it implies that, in case of exposures that are longer than this time, only a fraction of this will be with the crystal in the beam, thus more

background will be recorded. With short pulses (90 μs in the case of results reported in this manuscript), the crystal is in the beam for the whole 90 μs exposure.

Line 251/252 – Pulsed beam allows areas exposed to x-rays to be spaced out, also normally happen – see above.

R: Again we are not sure we correctly understand this comment. With shutterless acquisition and a continuous beam (i.e. the exposure time = period) the beam is constantly illuminating the jet, while on ID29 the pulsed beam produces an exposure time (90 μs) which is much shorter than the period (4.4 ms), making sure that between exposures the sample is fully replaced.

Line 252/253 – seems to play important role in mitigation of radiation damage, ensuring each diffraction image is collected from an unexposed area – also depends on speed.

R: Please see our previous comment.

Line 256/257 – needs a comparison to conventional microfocus beamlines with similar support and mother liquor. Need evidence of no cross-contamination.

R: This is what we show by collecting data from foils at different spacing (see Fig.3 and Results). Our findings on the sample we studied in the experimental conditions presented show that there is no cross-contamination. We cannot exclude that different experimental conditions or samples could show variation however. In regards to the point of comparing conventional microfocus beamlines with $\text{S}\mu\text{X}$, we agree with the statement but this is beyond the scope of this manuscript and will surely be the subject of future studies.

Line 258 – flux density achievable at ID29 places it in completely unexplored territory? What does this mean?

R: As we discuss in the following sentence, the flux density is a novelty in synchrotron beamlines and yet much lower than XFEL sources. For this reason, experiments with a similar flux density were, to the best of our knowledge, never performed before. We have modified this sentence.

Line 258/259 - 9 orders of magnitude lower than FELs and 3 orders higher than 3rd gen synchrotrons and other MX beamlines at next gen light sources? – Need actual values here. Also PETRA is 10^{13} so it is not 3 orders higher.

R: The values of flux and beam size are reported multiple times in the manuscript. For sake of clarity we have also explicitly added the flux density here. The flux density from other sources is extrapolated from the reported flux and minimum beam size although in some cases it can be overestimated by the assumption of a full acceptance of the focusing optics. In order to avoid any mistake we are now reporting “two to three orders higher than 3rd generation synchrotrons”.

Lines 260-264 - Why are the calculated doses much higher? Also, if the effect of radiation in RT crystallography is not well-characterised, why not discuss it here?

R: We are not sure we understand the first question. The doses were calculated as described in the Material and Methods using RADDPOSE-3D and they are that high because of high flux density and small crystal sizes. Moreover, based on the reviewer's comment we have verified our calculations and updated the results with the correct number of monomers in the unit cell. Radiation damage has been a matter of study in protein crystallography for more than 60 years and it is still a matter of investigation, in particular after the advent of new X-ray sources that are able to produce beam intensities that were not achievable before. In our manuscript we report that, despite using a dose much higher than the limit reported in previous experiments, we could not observe evident signs of radiation damage. We speculate that this might also be an effect of the high dose rate, which is much higher than what is achievable at other synchrotron sources, and that has been hypothesized in previous works (<https://doi.org/10.1107/S0907444912012553>). We are confident that this observation will stimulate upcoming new research projects in the field.

Line 281 – comparable throughput to conventional MX – the ability to do 100 structures in 10 hours is well below the throughput of conventional MX beamlines, which can easily achieve 25 – 30 samples per hour, and this number will increase dramatically as synchrotrons are upgraded.

R: From our experience in cryo-MX, achieving 20 samples/hour is a more realistic estimate (although at times it compromises quality over throughput), with the time-limiting step often the sample unloading/loading sequence. This would correspond to a throughput of 100 samples / 5 hours which is at a comparable scale of what we envisage (100 samples / 10 hours) for future room temperature structure-based drug discovery projects. This approach would also have the additional advantage of not requiring manual soaking and harvesting and provide a more physiological description of the protein active site (as reported in the manuscript introduction).

Line 284 - One could envisage reaching the throughput of up to 100 ligand-soaked data sets in a 10 hour beamtime with SuX. Based on numbers provided this does not seem feasible, elaborate or remove?

R: We have elaborated more on our calculation and completed the text (lines 301-302).

Line 292/293 – it is possible but are you achieving the same spatial resolution and structural quality?

R: This is what has been reported by many of ID29 users, the reference cited reports a structure that has been determined both at ID29 and at XFELs, leading to similar resolution and quality (albeit with lower sample consumption).

Calculation of dose – line 404 to 422 – need to provide raddose inputs, for example in the supplementary section. Absorbed dose should be in Gy not J/kg

R: All parameters used in RADDOSE-3D are reported in Supplementary Table 5 and we have added the scripts used. We have also included both Gy and J/Kg units.

Table 1 seems to be a less informative repeat of the other tables. Why is the data from the extruder the best (resolution, R_{work}, R_{free} etc)? Is it because more images were collected? Or different configuration of the beamline to allow a closer detector distance?

R: Table 1 is provided to give a summary on the data quality from SμX experiments presented in the manuscript. We have modified the title of Table1 for clarification. In regards to the extruders data, indeed more images were collected and Thaumatin crystals were larger than the other samples reported in the manuscript.

Supplementary table 2

What is the explanation for a high number of hits but the low indexing rate of those hits? What threshold was used to define a hit?

R: The hits were initially identified with a very conservative approach, leading to overestimation. The parameters chosen for the hit-finding are already mentioned in the material and methods. We used conservative settings for the spotfinding in order to not miss any potential information, considering that the beam divergence and bandwidth were a novelty.

CC* is normally put to 0.7 (or CC1/2 out to 0.3, which is not included). Statistics in table for multiple datasets go past this value, so the resolution cutoff should in fact be greater than 2.05. Plots of these values would help to support this

R: In serial crystallography processing CC* is more often reported instead of CC1/2. Moreover, there are no definite threshold for CC* cutoff, several papers reported structure with CC* lower than 0.7 (<https://www.nature.com/articles/s41467-017-00630-4>, <https://www.nature.com/articles/s41467-023-41246-1>, <https://www.nature.com/articles/s41467-023-43523-5>, <https://www.nature.com/articles/s42004-024-01236-w> to name some). Finally, the resolution cutoff is a combination on CC*, SNR, multiplicity and other parameters. We report the plot of the CC* statistics here below.

Average dose per crystal is very high, which is mentioned but not justified. Dose above 100 kGy often considered in literature as 'damaged'

R: This is indeed one of the results and observations of our manuscript. The high dose is calculated with RADDOSE-3D which is based on a subset of input parameters, such as flux (ph/s), exposure time (90 μ s) and beamsize (4 x 2 μ m), which on paper gives the high output value mentioned. However this dose is delivered on a much shorter time scale than previously possible in a pulsed and not in a continuous beam. RADDOSE does not take this into account during its calculations, with ID29 differentiating from other synchrotron beamlines, as mentioned. RADDOSE-XFEL would be an obvious alternative to try, as it calculates doses within a timed X-ray pulse but as mentioned in the main text, doses could not be calculated on the microsecond time scale using this software, which only assumes femtoseconds in its calculations. We, however, provide all the details of our calculations in Supplementary table 5 and now also include the RADDOSE-3D scripts so the reader knows where the values are coming from.

B factors for HEWL si chip and A2AR_Istradefylline is very high.

R: For HEWL we reported an average B-factor of 48.08 \AA^2 , which is not particularly high considering that the structure were determined from microcrystals at room temperature. In fact similar values were reported in previous works, such as in Stellato et al. (<https://doi.org/10.1107/S2052252514010070>), where an average B-factor of 51.7 \AA^2 for lysozyme at similar resolution. The same applies to A_{2A}R, where the intracellular domain is intrinsically flexible and B-factors as high as 129 \AA^2

(<https://doi.org/10.1107/S1600576723006428>) and 80 \AA^2 (<https://doi.org/10.21203/rs.3.rs-3994449/v1>) were reported in other $A_{2A}R$ structures.

Need beam size, flux and raddose scripts for these experiments.

Why is the detector always set to 150 mm? Should HEWL be able to get resolution down to ~ 1.7 ? Why so low?

R: The resolution was limited as the sample-to-detector distance was fixed to 150 mm due to technical limitation at the time of the experiments. This setup gave around 2.1 \AA on the edge explaining the resolution limits. The other parameters and scripts are now provided in the Supplementary materials.

Supplementary figure 2 – graph intensity should be in photons not ADU – this applies in multiple papers.

R: We agree with the reviewer that since the advent of the photon counting detector this has become the most common unit for intensities, however the JUNGFRU is an integrating detector in which images are recorded in 32 bits (see also <https://doi.org/10.1038/s41592-018-0143-7>). Since Supplementary Figure 2 is directly calculated from the images we prefer not to convert this to photons, but rather leave it in ADUs.

Supplementary Table 3 –

The number of collected images is much higher than chip datasets; why?

CC* for HVE_SACLA is 56.98 at the cutoff, why?

The average dose again seems very high

Plots comparing the statistics in addition to the maps would help to support this.

R: The difference is due to the fact that from a single Si-chip it is only possible to collect 25600 images and thus, for each condition, 2 chips were used to obtain a complete dataset. Viscous injectors were loaded with around $50 \mu\text{L}$ of sample, which can be extruded continuously for around 30 minutes, hence up to 500,000 images were collected. The higher redundancy obtained explains the high CC* values.

Supplementary table 4 –

IS the CC* listed the overall CC*? Why have they not included the resolution at the cutoff?

Completeness is low for $A_{2A}R$ dataset.

How is this processed? Have you tried using autofit in Phenix suite to see when fitting becomes impossible?

Resolution of each dataset? Fig 4. – do they really have the same resolution cutoff?

Plots of statistics please

R: Indeed the CC* is the overall one. The goal of this table is to give the overall statistics for every merged sub-datasets using the same resolution cutoff as the complete dataset. The completeness of $A_{2A}R$ data is reduced because of lower symmetry space group (when

compared to HEWL) and possibly a predominant orientation. As mentioned above we tried the phenix.ligandfit with success with the 2000 frames datasets which we identified as the minimal dataset for ligand binding determination (lines 234-236).

Supplementary table 5

What is beam size? Are you assuming top hat or gaussian? Did you change the beam size depending on the crystal size? Because if the beam size for the Injector is smaller, then the average dose per crystal (whole) and (exposed region) should be different

R: All the parameters used for the dose calculation are mentioned in the table. The beam has a gaussian profile and the beam size is fixed and is the same for all experiments.

Reviewer #3 (Remarks to the Author):

The manuscript describes the impressive technical work on the ID29 beamline at the ESRF. This instrument is pushing back the frontiers of synchrotron serial crystallography. The manuscript is very well written, cites the literature adequately and contains sufficient technical information to describe the work.

R: We thank the reviewer for the positive feedback and the appreciation of our work

I have a few minor points for correction:

1. Title "... microseconds X-ray pulses ..." should be changed to "... microsecond X-ray pulses ...".

R: We corrected this.

2. Line 388, Data collection parameters; include value of flux at sample.

R: We added the maximum flux. When using attenuated beam, the value is explicitly reported in Supplementary Table 5

3. Line 401, "... 5WR880, and 5NM214 respectively." Should be changed to "... 5WR880 and 5NM214, respectively."

R: We corrected this.

4. Lines 683-684: "The absence of "lunes" indicates that no rocking takes place during the acquisition." Lunes are visible on the left-hand side of Fig2a? Consider re-phrasing or delete the sentence.

R: We agree with the reviewer that we used the term "lunes" improperly, we have removed this sentence.

5. Lines 181-185 and Fig.3. The difference in observed scattering and B-factors between the 20 micron spacing data and the 10/50 micron data are non-trivial. Is this the result of sample batch variation? The higher scattering and lower B-factors for the 20 micron dataset should be mentioned in the results section of the manuscript.

R: Indeed we consider the difference an effect of batch to batch variation, but also the 20 micron option is the best combination to maximize the data that can be collected from a single support while maintaining a safe spacing. We have added a comment in the manuscript.

6. Supplementary Fig3. Panels a,b & c need to be shown at the same magnification as panel d. It is not possible to assess crystal size from panels b and c at such a low magnification.

R: In fact these photos were selected to illustrate the density of the samples used. Following the reviewer's suggestion, we have replaced the photos with higher magnification versions.

Apart from a few minor corrections the manuscript, and the work described, are excellent.

R: We thank the reviewer for such a positive evaluation.

Reviewer #4 (Remarks to the Author):

I co-reviewed this manuscript with one of the reviewers who provided the listed reports. This is part of the Communications Chemistry initiative to facilitate training in peer review and to provide appropriate recognition for Early Career Researchers who co-review manuscripts.

Reviewers' comments:

Reviewer #1 (Remarks to the Author):

I appreciate the hard work the authors have put into revising the manuscript. They have comprehensively addressed the points raised during the previous review, and the modifications made have improved the clarity and quality of the paper. I am satisfied with the changes and have no further comments. I recommend that the manuscript be accepted for publication in its current form.

R: We thank the reviewer for the positive evaluation towards publication, and for the valuable feedback provided that contributed to improve the manuscript.

Reviewer #2 (Remarks to the Author):

Thank you for addressing our comments in such detail and making changes, including the calculations and the automated ligand fitting calculations. We feel these changes significantly strengthen the manuscript. However, we want to follow up on a few responses.

We acknowledge the comments regarding the restricted resolution limits for some datasets. While we understand that geometrical constraints were the limiting factor, we believe that recollecting the data at a higher resolution—if that is now feasible and not too time-consuming—would better showcase the impressive capabilities of the beamline. It may not be the only data quality indicator, but it is an important metric for most users. Repeating measurements is not necessarily required for publication of this manuscript, but a ~ 2.05 Å HEWL dataset may not be the best benchmark for your beamline.

R: We appreciate the reviewer's overall positive remarks on the "impressive capabilities of the beamline" thanks to the uniqueness of microsecond pulses in a larger bandwidth, which is one of the main messages of our manuscript. This allows us to obtain high quality data (as shown by the ligand electron density maps) with small sample consumption. We believe marginal improvement in the resolution using new data will not change these findings.

The phenix.ligandfit of the GlcNac substrate has a poor RMSD of 2.53Å, which is explained in the text by the rotation along the glucose C2-C5 direction. But could this also be shown in the

figure, i.e., a superposition of the ligand as found by phenix.ligandfit vs the correctly fitted ligand? This would show that the ligand fitting was mostly successful despite the large RMSD. Please could you amend the figure to include this?

R: We thank the reviewer for the suggestion and we agree that a figure would improve this result. We have added it as a new Supplementary Figure 3, to preserve the readability of Figure 4. We have also added a sentence (lines 214-215) to describe this.

Thank you for including the CC* plots in your response to us. Could these also be included in the paper? We think this would be informative to readers. Also, please plot these against a linear scale such as $1/d$ so they can be viewed more clearly. There is indeed no definitive 'correct' threshold for CC*, which is why showing the plots is informative. CC* is not more common for serial crystallography data, it is a measure of the potential quality of the dataset, both CC* and CC $\frac{1}{2}$ are equally valid. From your CC* plots, some of your data seem to have a low correlation in the low-resolution shells. Please could you also include in the manuscript the peakogram plot from crystfel to show you have not overloaded your detector; this is a common reason for this pathology. If it turns out you do have overloads, perhaps this would be another good reason to recollect the data.

R: We think that including this plot in the manuscript, even in supplementary, would not add much relevant information as we are not comparing data quality across different beamlines or using different data reduction pipelines. Additionally, we included the peakogram plots below to confirm that there is no detector saturation in the data reported. Moreover, we think that the peakogram plots we have included in this response would also not add any significance to the paper for the same reason. Hence we prefer not to include them in the manuscript.

You frequently state that data collected on ID29 is comparable to data collected at an XFEL, however, it is perhaps equally informative to see how these data compare to data collected at

3rd generation sources. For example, <https://pubs.aip.org/aip/rsi/article/90/8/085110/360302/The-serial-millisecond-crystallography-instrument>, was collected at the Australian Synchrotron. In this paper they collected Lysozyme data to 1.83 Å. This is by no means an exhaustive search. Still, if the goal of this paper is to demonstrate the benefits of ID29 over previous beamlines, then this may be another good reason to recollect your data, given it takes ~20 minutes to collect a full dataset (Table 1). If you want to compare your data to SFX data collected at XFELs, please include a genuine comparison of example data from an XFEL in the manuscript to support this statement. As we have stated, we would also advise making a similar comparison with SSX data from 3rd generation sources. There will obviously be differences in sample delivery method, etc, but these can be discussed in the manuscript. This will give the reader a better idea of where this beamline sits compared to its competitors.

R: For this point, we beg to differ with the reviewer on the frequent statement about comparison. We have instead referenced a paper (Grieco et al 2024) which report the structure of a real life example, the NQO1 protein, collected at ID29 (at 2.5 Å resolution) which had been previously collected at EuXFEL (Doppler et 2023, <https://doi.org/10.1039/D3LC00176H>), at the resolution of 2.7 Å, supporting our statement. However, the goal of the paper is to show the capabilities of the microsecond pulses with unique beam characteristics for serial crystallography, which complements rather than competes with other light sources. Moreover, the complementarity across facilities has already been discussed in the manuscript section.

We would like to try again to clearly state that the time a sample is exposed to the beam is dictated by a combination of the delivery method and the source. The manuscript currently implies that the exposure time of ID29, means that the crystal is continuously in the beam during an exposure, and that this could not occur at other sources. This is simply not true for all delivery methods. For example, jets can be run slower so crystals spend more time in the beam, and fixed targets can literally stop moving such that the crystal can be immobilized in the beam for as long an exposure time as required. Please rewrite this point, specifying that with ID29 jets can be run faster to achieve this goal or simply delete this point.

R: We thank the reviewer for this suggestion. For sake of clarity we now specify that this sentence is valid only when running at similar speed (line 265). However, we beg to disagree that a jet can be run at any speed. In the case of ID29, the exposure time is 100-1000 times shorter (90 μs) than at 3rd generation synchrotrons (> 1ms). Running a jet 1000 times slower to keep the crystal in the beam for the whole exposure, would make it impractical to jet in a stable way with many matrices.

While we can't account for this paper you mentioned in your rebuttal, (Zhao, et al 2019. "A Guide to Sample Delivery Systems for Serial Crystallography." The FEBS Journal 286 (22): 4402–17. <https://doi.org/10.1111/febs.15099>), but can you state reasons in the manuscript as to why, with the same sample delivery method, your consumption is less than an XFEL? Assuming you optimized your rate of sample delivery to your source, there should be no difference in sample consumption at an XFEL, a 4th generation source or a 3rd generation source. The only

thing that changes is the rate at which you will acquire data and, perhaps, the stability of your jet. It is true, when you introduce a laser pump pulse into an experiment, this will change the required jet speed. If this is the point you wish to make, you need to mention the trade-offs between pump pulse and jet speed.

R: We believe that in the manuscript we clearly state already that there are different reasons for a more efficient sample consumption (and we report the case of $A_{2A}R$ structure obtained using only 15 μ l of LCP bolus). As we mentioned above, not all sample delivery methods can be adapted to any repetition rate, hence lower repetition rate sources (<100 hz as most XFELs and 3rd generation synchrotrons) intrinsically implies larger sample consumption. However, another important aspect that is not considered in the reviewer's comment is the fact that ID29 presents a larger bandwidth and we show that as little as 2000 merged frames are sufficient to obtain details of a ligand in the electron density map. The need of recording less frames implies the need of less sample thus also a lower sample consumption, and this is already addressed in the manuscript.

Please plot radial integrals in photon counts or ADUs consistently throughout your manuscript. Preferably in photon counts as this is more recognizable across more detector types.

R: We agree on the need for consistency, however we prefer to keep the units in ADUs, because the detector is not a photon counting one and the raw data are recorded in ADUs. For this reason we have updated Figure 3 labels and legends accordingly. Moreover, keeping these figures in ADUs will maintain the consistency with the raw data that will be deposited in the ESRF data portal. We have specified in the legend the equivalence of photons to ADUs for the sake of completeness.

Reviewer #3 (Remarks to the Author):

The revised manuscript has addressed this reviewers corrections and is suitable for publication.

R: We thank reviewer #3 for their valuable feedback and positive evaluation of the manuscript towards publication.

Reviewer #4 (Remarks to the Author):

I co-reviewed this manuscript with one of the reviewers who provided the listed reports. This is part of the Communications Chemistry initiative to facilitate training in peer review and to provide appropriate recognition for Early Career Researchers who co-review manuscripts.

Dear Dr Huijuan Guo

We once again would like to express our sincere gratitude to you and the four reviewers for the thorough review and insightful comments on our manuscript titled "Advancing macromolecular structure determination with microseconds X-ray pulses at a 4th generation synchrotron" (COMMSCHEM-24-0459A).

We enclose a final amended version of the manuscript following the Editorial Requests Table instructions.

With best regards

Daniele de Sanctis